# DeltaEvolve: Accelerating Scientific Discovery through Momentum-Driven Evolution

**Jiachen Jiang** [1]  **Tianyu Ding** [2]  **Zhihui Zhu** [1]

## Abstract

LLM–driven evolutionary systems have shown promise for automated science discovery, yet existing approaches such as AlphaEvolve rely on full-code histories that are context-inefficient and potentially provide weak evolutionary guidance. In this work, we first formalize the evolutionary agents as a general Expectation–Maximization framework, where the language model samples candidate programs (E-step) and the system updates the control context based on evaluation feedback (M-step). Under this view, constructing context via full-code snapshots constitutes a suboptimal M-step, as redundant implement details dilutes core algorithmic ideas, making it difficult to provide clear inspirations for evolution. To address this, we propose `DeltaEvolve`, a momentum-driven evolutionary framework that replaces full-code history with structured `semantic delta` capturing how and why modifications between successive nodes affect performance. As programs are often decomposable, `semantic delta` usually contains many effective components which are transferable and more informative to drive improvement. By organizing `semantic delta` through multi-level database and progressive disclosure mechanism, input tokens are further reduced. Empirical evaluations on tasks across diverse scientific domains show that our framework can discover better solution with less token consumption over full-code-based evolutionary agents.

## 1. Introduction

LLMs have facilitated automated science discovery across diverse domains such as mathematical optimiza-

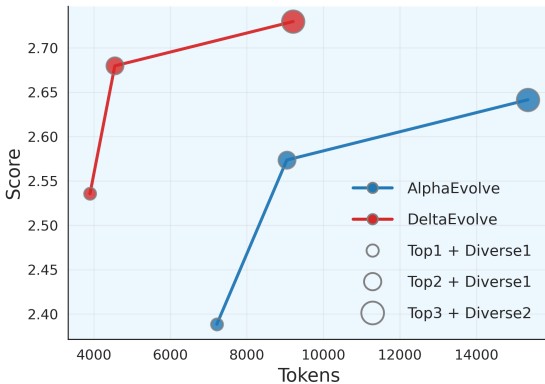

*Figure 1.* Comparison between AlphaEvolve and `DeltaEvolve` (ours) on black-box optimization over 100 iterations. The x-axis shows cumulative consumed tokens across all iterations, and y-axis shows the best achieved score. Point size indicates the number of top and diverse nodes included in the context. `DeltaEvolve` consistently achieves higher scores with fewer tokens.

tion(Georgiev et al., 2025; Hubert et al., 2025; Trinh et al., 2024), physical systems (Li et al., 2025), and molecular discovery (Averly et al., 2025). Although different tasks, the common core objective is to discover a high-performing object that satisfy desirable quantitative properties. Fundamentally, all these tasks can be represented as the search problem: solutions are difficult to synthesize directly but computationally easy to evaluate. Recent works (Real et al., 2020; Mankowitz et al., 2023) mainly choose to search in the code space, since code is sufficiently expressive to represent complex algorithms, aligns well with LLM pretraining, and enables automatic evaluation through execution.

For simple problems, single-turn generation conditioned on problem specifications can already be effective. However, complex tasks typically require interaction with execution environments and iterative refinement based on feedback. This has led to self-evolving agents such as FunSearch (Romera-Paredes et al., 2024) and AlphaEvolve (Novikov et al., 2025), which iteratively generate programs, execute them, analyze feedback signals, and produce improved variants. These approaches can be interpreted as a form of test-time scaling, instead of relying solely on internal reasoning tokens, the performance improves by allocating more computation to accumulating history and feedback across iterations.

[1]Department of Computer Science and Engineering, The Ohio State University, Columbus, OH, USA [2]Microsoft, Redmond, WA, USA. Correspondence to: Zhihui Zhu <zhu.3440@osu.edu>.

*Proceedings of the 43$^{rd}$ International Conference on Machine Learning*, Seoul, South Korea. PMLR 306, 2026. Copyright 2026 by the author(s).

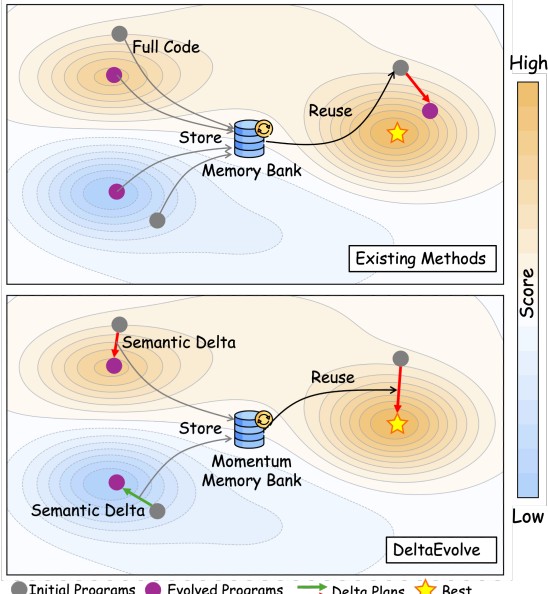

*Figure 2.* Illustration of search dynamics under existing methods and `DeltaEvolve`. Existing methods store and reuse full programs, whereas `DeltaEvolve` stores `semantic delta` that capture what changed and why it worked, forming a momentum-like memory that provides more informative guidance for reuse.

Despite their success, existing evolving agents face two fundamental limitations:

- **Limited context window.** Extensive multi-turn inference incurs prohibitive computational cost and latency, preventing effective utilization of the full evolution history (Bansal et al., 2025). As a result, methods such as AlphaEvolve (Novikov et al., 2025) retain only a small subset of high-quality solutions (e.g., top-performing or diverse programs). However, for complex systems, individual programs are already long, leaving limited capacity to reuse historical information.

- **Insufficient evolutionary guidance.** Complete programs often entangles core algorithmic ideas with extensive implementation and control logic which are irrelevant to the underlying strategy, making it hard for LLMs to isolate the truly useful components from vast code histories. Therefore, full codes would not explicitly capture transferable patterns of success or failure that could guide future iterations, causing missed successful patterns or repeated failures.

This motivates our research question: *how can we provide stronger evolutionary guidance while staying within a limited context budget?* To rigorously address this, we first formalize the evolutionary agents within a general Expectation–Maximization (EM) framework. Optimization proceeds by alternating two steps: in the E-step, the language model samples candidate programs conditioned on the current context; in the M-step, the system updates the control

context using evaluation feedback to maximize the objective. From this perspective, the standard practice of representing history via static full-code snapshots constitutes a suboptimal M-step: while it captures the current state, it obscures which changes led to performance gains or losses, weakening the learning signal.

We hypothesize that within a fixed task, the modifications that affect performance in earlier iterations often capture task-level structures and may remain informative for guiding updates to different programs in later iterations. These semantic changes are more informative for guiding future updates than the static solutions themselves. Based on this hypothesis, we propose `DeltaEvolve`, a momentum-driven evolutionary framework designed to optimize the M-step. As illustrated in Figure 2, instead of storing full-code snapshots in the memory bank, we propose to use `semantic delta` to record the core semantic modifications from a parent node to its offspring and their resulting impact on performance. The memory bank stores `semantic delta` between pairs of nodes. As these deltas capture what changed across successive iterations and guide future updates, they form a directional signal analogous to the *momentum* term in optimization.

To implement `semantic delta` efficiently, we augment each node with additional structures: a delta summary (Level-1) and delta plan details (Level-2), alongside the existing full code (Level 3), forming a pyramid-structured `Multi-Level Database`. Then the `Progressive Disclosure Sampler` dynamically decides the level of detail to expose for historical nodes based on relevance and recency: the current parent node is presented as full code for editing, while historical inspiration nodes are compressed into delta summaries or delta plan details. This approach exposes the logic of past improvements without wasting tokens on their implementation details.

Our evaluation prioritize two critical metrics: solution quality and token consumption. As illustrated in Figure 1, on the black-box optimization task, `DeltaEvolve` consistently achieves higher scores with significantly fewer input tokens across varying context hyperparameters (e.g., number of top and diverse programs). To ensure generality and robustness, we further validate our framework across five diverse domains. Extensive experiments confirm that `DeltaEvolve` discovers solutions of comparable or superior quality to state-of-the-art baselines, all while reducing total token consumption by approximately 36.79% on average.

Overall, our work makes the following contributions:

- **General Evolutionary Framework.** We formalize the evolution as an Expectation–Maximization (EM) process, identifying updating the context (the M-Step) as the critical bottleneck in existing systems.

- **DeltaEvolve.** We propose a momentum-driven framework that replaces static full-code snapshots with `semantic delta`. This approach utilizes a progressive disclosure mechanism to maximize context utility.
- **Empirical Efficiency**: Through comprehensive evaluations across five scientific domains, we demonstrate that `DeltaEvolve` achieves superior solution quality while reducing token consumption compared to state-of-the-art full-code baselines.

## 2. Related Works

**Evolutionary Coding Agents.** The integration of LLMs with evolutionary algorithms has produced powerful frameworks like AlphaEvolve (Novikov et al., 2025) and OpenEvolve (Sharma, 2025), which established the core viability of LLM-driven program evolution. Subsequent works have significantly advanced LLM-driven evolution through improved orchestration. Building on this foundation, subsequent works have focused on optimizing the orchestration of the search process. Specifically, ShinkaEvolve (Lange et al., 2025), GigaEvo (Khrulkov et al., 2025), and CodeEvolve (Assumpção et al., 2025) introduce advanced population management techniques, including novelty-based rejection sampling, MAP-Elites quality-diversity algorithms, and island-based crossover operators. Moreover, ThetaEvolve (Wang et al., 2025b) explores fine-tuning smaller LLMs via Reinforcement Learning. However, despite these unique optimization strategies, all these systems fundamentally rely on representing historical solutions as full source code within the context window. This approach imposes a rigid token bottleneck that forces a trade-off between history length and detail; in contrast, `DeltaEvolve` resolves this by representing history as `semantic delta`, enabling the agent to evolve with minimal token cost.

**Context Engineering.** Recent work on agent prompting and context engineering has investigated how to structure and compress context to support long-horizon reasoning and iterative agent behavior. Practical agent systems and guidelines (Anthropic, 2025) emphasize modular prompts and progressive disclosure, while memory-augmented agents such as MemoryLLM (Wang et al., 2024) and its scalable extension M+ (Wang et al., 2025a) introduce long-term memory to mitigate context overflow in extended interactions. In parallel, prompt and context compression methods (Zhang et al., 2024; Fei et al., 2025; Shi et al., 2025) aim to reduce effective context length through token selection or adaptive refinement while preserving task-relevant information. However, these existing techniques are inherently generic; they compress information without awareness of optimization objectives. In contrast, our work does not merely reduce context length, but actively constructs context that directs the agent toward higher-quality solutions.

## 3. Evolutionary Framework

In this section, we aim to examine LLM-driven evolution from a rigorous optimization perspective.

### 3.1. Problem Definition

We consider a broad class of open-ended problems arising in mathematics, scientific discovery, and engineering. These tasks include, but are not limited to, classical optimization problems, algorithm and code design, combinatorial constructions, heuristic discovery, and scientific or engineering workflows where the solution structure or algorithmic strategy is not known a priori and must be discovered through interaction with an evaluator, rather than optimized within a fixed parameterization.

In this work, we focus on problems whose solutions, or the procedures for discovering solutions, can be expressed as programs. Programs provide a flexible and executable representation that supports structured modification, composition, and verification, making them particularly well suited for iterative improvement. But we note that the same principles apply to other forms of solution representations (such as symbolic constructions or mathematical proofs) provided they admit an evaluator that offers task-dependent feedback. Programs serve here as a concrete instantiation that enables precise execution and scalable evaluation.

Formally, let $\mathcal{P}$ denote the space of programs. For a given task or problem $q$, let $R_q : \mathcal{P} \to \mathbb{R}$ be an evaluator that assigns a scalar score or reward (e.g., accuracy, runtime, objective value, or combinations) to each program by executing it. The goal is to find a program $p^\star$ that acheives the highest possible score, i.e.,

$$p^* = \arg \max_{p \in \mathcal{P}} R_q(p), \tag{1}$$

Although (1) resembles a conventional optimization objective, it differs from classical mathematical optimization in several fundamental ways: $(i)$ the task specification $q$ may be partially formal and partially expressed in natural language, making the objective implicitly defined through the evaluator rather than an explicit analytic form, $(ii)$ the search space $\mathcal{P}$ consists of programs with variable length, control flow, and data structures, resulting in a discrete, extremely large space without a fixed dimensionality or natural notion of locality as in $\mathbb{R}^d$ space, $(iii)$ the evaluator $R_q$ is procedural and non-differentiable, often involving program execution and hard correctness constraints, which together induce a highly irregular and discontinuous objective landscape, and $(iv)$ candidate programs may themselves encode search or optimization procedures, so the problem amounts to optimizing processes that perform optimization internally. These characteristics place (1) outside the scope of classical optimization methods and motivate search-based, feedback-

driven approaches such as AlphaEvolve. Example in Table 3 further illustrates the difference.

## 3.2. The EM Interpretation

Since the objective function[1] $R$ is the only accessible feedback mechanisms (analytical gradients $\nabla R$ are inaccessible), the discovery task falls under the domain of zero-order black-box optimization. A standard approach for such problems is to employ an iterative Expectation-Maximization (EM) strategy: first, sampling from a distribution model(e.g. multivariate normal distribution) (Hansen, 2016) with latent variables (E-Step), and subsequently updating the latent variables based on the observed evaluation (M-Step).

However, the reward landscape in scientific discovery is exceptionally sparse and complex, rendering traditional surrogate models insufficient. Therefore, current agents increasingly leverage the reasoning capabilities of LLMs as a powerful distribution model. Its weight parameters and context would serve as the latent variables. We use $\mathcal{A}_\theta(q \oplus \mathcal{C})$ to denote an LLM that returns a likely program $p$ given the problem description $q$ and additional context $\mathcal{C}$, where $q \oplus \mathcal{C}$ denotes the concatenation of the two text inputs. In this paper, to accommodate the use of frontier LLMs, we assume that the model parameters $\theta$ are fixed and that only black-box access via an LLM API is available. We therefore treat the context $\mathcal{C}$ as a latent variable to be optimized. Under this formulation, we solve the objective in (1) via

$$\max_{\mathcal{C}} R(\mathcal{A}_\theta(q \oplus \mathcal{C})). \qquad (2)$$

Although it appears similar to prompt optimization or engineering (Ramnath et al., 2025), (2) differs fundamentally in that it focuses on *problem-specific context optimization*. Rather than learning a generic prompt that improves average performance across tasks, the context $\mathcal{C}$ is tailored to a given problem instance $q$ and optimized to maximize the objective $R$, acting as a *latent*, adaptive variable rather than a reusable instruction template. Intuitively, conditioning on $\mathcal{C} = p^\star$ increases the likelihood of generating $p^\star$ or related high-quality programs, whereas poor or random contexts degrade performance. *From a Bayesian perspective, $\mathcal{C}$ functions as an inductive bias over programs, with more informative contexts shaping the conditional distribution of the LLM toward higher-reward solutions.* However, directly optimizing $\mathcal{C}$ is challenging due to the high-dimensional, discrete, and non-convex objective. We therefore adopt an iterative optimization strategy inspired by the expectation–maximization (EM) framework, which alternates between refining the distribution over candidate programs and updating context to increase expected reward.

---

[1]To simplify the notation, when it is clear from the context, we will simply denote the evaluator $R_q$ via $R$.

1. **E-Step (Sampling).** Given the current context $\mathcal{C}_t$, the system samples candidate programs to estimate the local landscape. We also query the evaluator to obtain the history $\mathcal{H}_{\text{new}}$:

$$\mathcal{H}_{\text{new}} = \{(p_t, R(p_t)) \mid p_t = \mathcal{A}_\theta(q \oplus \mathcal{C})\}. \qquad (3)$$

2. **M-Step (Context Update).** The system updates the context to $\mathcal{C}_{t+1}$ to maximize the expected score in the next iteration. Using the accumulated history $\mathcal{H}_{t+1} = \mathcal{H}_t \cup \mathcal{H}_{\text{new}}$, also referred to as the memory bank, we define the update via a policy $\pi$:

$$\mathcal{C}_{t+1} = \pi(\mathcal{H}_{t+1}). \qquad (4)$$

It reveals that the policy of constructing context from history plays a key role in updating the search distribution and guiding evolution.

This framework provides a unified perspective on system improvement. In this setting, agents typically assume fixed model weights[2], thereby focusing the optimization on the M-Step: designing the policy $\pi$ to construct the most effective context from history. A naive strategy, such as greedy refinement, builds the context using a single program, such as the most recent or the highest-performing one. Conditioned on this program, the LLM attempts to modify it to better solve the target problem; we refer to this program as the parent program $p_{\text{parent}}$. This procedure is analogous to classical hill-climbing in descent-based optimization methods such as gradient descent. However, when conditioned solely on the parent program, the agent is susceptible to *mode collapse*, in which it repeatedly produces similar modifications across iterations, thereby limiting exploration and hindering escape from local optima.

Evolutionary system like AlphaEvolve (Novikov et al., 2025) attempt to enhance this by including $n$ "inspiration nodes" in the context as,

$$\mathcal{C}_{t+1} = \{p_{\text{parent}} \oplus \underbrace{\{(p_1, R_1), \ldots, (p_n, R_n)\}}_{\text{Full Code Inspirations}}\}. \qquad (5)$$

Each node stores the full program together with its evaluation score returned by the evaluator. The set of inspiration nodes typically consists of Top-$k$ and Diverse-$m$ programs. Top-$k$ nodes are selected based on the evaluator feedback $R$, while Diverse-$m$ nodes are chosen according to the semantic similarity of code texts. These inspiration programs provide additional contextual signals that guide program modification, analogous to leveraging neighboring points to estimate a descent direction in gradient-based optimization.

---

[2]Updating $\theta$ corresponds to E-Step optimization, which has been explored in very recent works like ThetaEvolve (Wang et al., 2025b) and TTT-Discover (Yuksekgonul et al., 2026).

**Significance of Framework.** Evolving systems are traditionally viewed through the lens of agentic interaction—similar to reinforcement learning (RL)—where an agent maximizes reward through feedback. However, unlike RL, these systems cannot perform parameter updates, leaving the mechanisms of convergence and "learning" theoretically ambiguous. By formalizing evolution as a blackbox optimization problem within an EM framework, our framework isolates the true driver of the system. It reveals that in the absence of weight updates, the context $\mathcal{C}$ acts as the sole proxy for learned variables. Consequently, the M-step plays a role analogous to a gradient update, a connection we exploit in the next section to develop more efficient optimization strategies.

### 3.3. Context Selection Dominates Scalar Feedback

While evaluator feedback $R$ serves as the directional signal for evolution, the precise mechanism driving the optimization remains unclear. Our framework reveals that the objective $R$ serves a dual role in the M-Step:

1. **Context Selection.** Guiding selection policy $\pi$ to choose high-quality codes from memory bank into context window $\mathcal{C}$ (e.g., filtering high-quality solutions);

2. **Scalar Condition.** Providing numerical feedback correlates to the code directly within $\mathcal{C}$ (e.g., "Score: 0.95") to condition the LLM's next generation.

This duality raises a fundamental question regarding the nature of LLM-based optimization: *Does the performance gain arise from mimicking the high-quality solutions preserved by the selection policy, or from correlating code with numerical scores?* To answer this question, we decouple their effects and design the following controlled experiments.

**Experimental Design.** We conduct a controlled study on problems from five domains using the AlphaEvolve framework. We compare three distinct settings:

- **AlphaEvolve (Standard):** Uses Top-$k$ selection ($\pi_{\text{elite}}$) and includes explicit scores in the prompt.

- **Blind-Elite:** Uses Top-$k$ selection but **masks** all numerical scores from the prompt. The LLM receives the best codes but not the values.

- **Random-Context:** Randomly select the historical programs ($\pi_{\text{rand}}$) from the histories but including scores.

**Results & Insight.** As shown in Table 1, the results are revealing. Removing numerical scores (*Blind-Elite*) results in similar performance compared to the AlphaEvolve. In contrast, removing the selection policy (*Random-Context*) causes performance to collapse, even when scores are clearly visible.

*Table 1.* Ablation study on evaluator feedback mechanism (using `gpt-5-mini/o3-mini`). We compare `AlphaEvolve` (Standard), `Blind-Elite` (No Score), and `Random-Context` (No Selection) using objective scores.

| Task | AlphaEvolve (Standard) | Blind-Elite (No Score) | Random-Context (No Selection) |
|---|---|---|---|
| 1. Blackbox Optimization | **2.642** | 2.578 | 1.429 |
| 2. Hexagon Packing | **0.972** | 0.970 | 0.786 |
| 3. Symbolic Regression | **3.265** | 3.179 | 2.576 |
| 4. PDE Solver | **0.885** | 0.884 | 0.803 |
| 5. Efficient Convolution | 0.897 | **0.911** | 0.550 |

**Implication** These findings suggest that standard evolutionary agents operate primarily via in-context learning, not implicit regression. The agent improves by utilizing patterns from the high-quality context selected by the evaluator. The scalar feedback fails to guide the LLM because it provides a magnitude of success without explaining the mechanism of improvement.

## 4. DeltaEvolve

In this section, we first motivate replacing full-code with `semantic delta` (Section 4.1), then describe their implementation via a multi-level database (Section 4.2) and a progressive disclosure sampler (Section 4.3).

### 4.1. Motivation

Building on the framework in Section 3, we identify the `M-Step` as the decisive factor in agent design, as it governs the evolutionary trajectory. The core challenge lies in designing more effective strategy $\pi$ to construct the context $\mathcal{C}$ from history that effectively guides the search.

However, we argue implementing `M-step` using Equation (5) is still suboptimal: it populates $\mathcal{C}$ with full code snapshots and raw numerical scores. It provides a static *state description* rather than an *update gradient*. While it captures where the search is, it obscures the trajectory of how it got there.

To address this, we observe that in the domain of algorithmic discovery, programs are not monolithic entities; they are *decomposable* structures. The drivers of evolution are not entire solutions, but some effective components. These logical modifications are often transferable and more informative than the static solutions themselves. Based on this insight, we introduce `semantic delta` to explicitly capture these transferable components:

$$\delta_i = \text{Diff}(p_i, p_{i-1}). \tag{6}$$

Here, $\delta_i$ represents the interpretable logic change between a program and its parent. The $\delta_i$ usually consists of multi-

*Figure 3.* Comparison of the `DeltaEvolve` pipeline with AlphaEvolve. `DeltaEvolve` incorporates `semantic delta` into the context window instead of full code.

ple isolated modification components, and each component records the exact logic of which part is changed in concise natural-language description. Refer to Figure 14 in the appendix for an example of these decomposed components.

By accumulating `semantic delta` across different nodes, we construct the context of `DeltaEvolve` as,

$$\mathcal{C}_{\text{delta}} = \{p_{\text{parent}} \oplus \underbrace{(\delta_1, \Delta R_1), \dots, (\delta_n, \Delta R_n)}_{\text{Delta Inspirations}}\} \quad (7)$$

where $\Delta R$ denotes a qualitative performance shift (e.g., "Improved or Degraded") rather than a precise numerical change (e.g., "±0.15"). As analyzed in Section 3.3, exact numerical gains are often less informative and unnecessary.

This formulation closely aligns with the concept of momentum in classical optimization. Just as SGD methods accumulate gradients through successive differences between iterates $(x_i - x_{i-1})$, the `semantic delta` capture the differences between successive programs $(p_i, p_{i-1})$, In this sense, the accumulated deltas serve as a discrete, semantic analogue of a momentum vector, encoding the prevailing direction of improvement across iterations.

By eliminating redundant implementation details, this representation substantially improves token efficiency, enabling the model to attend to a longer and more informative history of algorithmic evolution. To further mitigate token overhead in practice, we show the implementation details of `semantic delta` in the following sections.

## 4.2. Multi-Level Database

To implement `semantic delta` efficiently, we augment each node with two additional structures: a delta summary (Level-1) and delta plan details (Level-2). Together with the existing full code (Level 3), each node in the database is represented as a pyramid structure:

$$N_t = (\delta_t^{(1)}, \ \delta_t^{(2)}, \ p_t), \quad (8)$$

where both $\delta_t^{(1)}$ and $\delta_t^{(2)}$ serve as the `semantic delta` representations—distinguished primarily by their token numbers—while $p_t$ represents the full executable code.

**L1: Delta Summary** $(\delta_t^{(1)})$. $\delta_t^{(1)}$ describes the overall strategy change from the parent node. It focuses on high-level ideas and avoids implementation details, providing a compact description of the search direction (typically 20–40 tokens). We use an explicit `FROM`/`TO` format:

$$\delta_t^{(1)} := \langle \texttt{FROM:} \ s_{t-1} \quad \texttt{TO:} \ s_t \rangle, \quad (9)$$

where $s_t$ is the high-level idea summary of the node $t$. Please refer to Figure 13 for an example.

**L2: Delta Plan Details** $(\delta_t^{(2)})$. $\delta_t^{(2)}$ is a structured plan consisting of $J$ multiple modifications. For each modification, we contrast the old logic with the new logic and state the underlying hypothesis. This representation makes the reason for improvement explicit, allowing the agent to learn what to change and why without reading the full code:

$$\delta_t^{(2)} := \{(\ell_{t-1}^{(j)}, \ell_t^{(j)}, \texttt{hyp}_t^{(j)})\}_{j=1}^J, \quad (10)$$

where $\ell_t^{(j)}$ denotes the $j$-th logic component of node $t$, and $\texttt{hyp}_t^{(j)}$ is the hypothesis predicted by the LLM to explain the modification. Please refer to Figure 14 for an example.

**L3: Full Code** $(p_t)$. $p_t$ stores the full program of node $N_t$ and remains essential for evolution, since node evaluation relies on executable code and new nodes must be derived by modifying the parent program rather than abstract plans.

All three levels are co-generated by the LLM during mutation. We enforce a strict output format requirements (as shown in Figure 5) that requires LLM generate all three levels in order. A lightweight parser extracts content between explicit delimiters (e.g., `#Delta-Summary-Start` and `#Delta-Summary-End`), enabling reliable retrieval of different parts.

## 4.3. Progressive Disclosure Sampler

To further reduce token consumption, we employ a progressive disclosure mechanism to control the level of detail

exposed in context. Rather than treating all historical nodes uniformly, the prompt sampler adaptively adjusts the abstraction level of each node based on its relevance and recency, revealing either coarse-grained summaries or finer-grained deltas as needed to guide the next mutation.

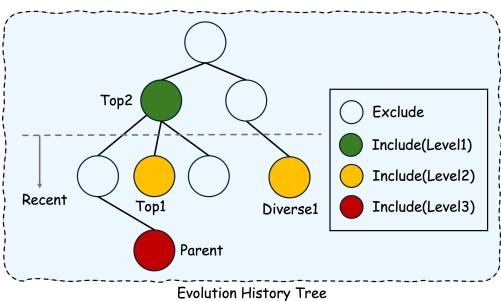

*Figure 4.* Progressive Disclosure Sampler

The sampler operates in two stages: (1) `Node Selection`, which selects historical nodes for inclusion, and (2) `Multi-Level Rendering`, which assigns an appropriate abstraction level (Level 1, 2, or 3) to each selected node.

**Phase 1: Node Selection.** Given the database $\mathcal{D}$ at step $t$, we select three distinct sets of nodes to construct the context:

1. **Parent Node** ($N_{\text{parent}}$): The solution selected for modification. The parent is sampled using a stochastic selection policy $\pi_{\text{select}}$ that prioritizes high-reward nodes while retaining a non-zero probability of selecting lower-reward ones, thereby balancing exploitation and exploration.
2. **Elite Nodes** ($\mathcal{N}_{\text{top}}$): A set of the $k$ high-scoring solutions discovered so far. These nodes represent successful strategies and capture the dominant directions of optimization.
3. **Diverse Nodes** ($\mathcal{N}_{\text{div}}$): To mitigate mode collapse, we sample $m$ diverse nodes based on the similarity of their text-embedding representations. Specifically, nodes are organized into a MAP-Elites grid[3], and candidates are selected from cells that are maximally distant from the parent, thereby injecting novel logic into the prompt.

**Phase 2: Progressive Rendering.** After selecting the relevant nodes, we render them into the context with different levels of detail:

- **Level 1 (Ancient History).** For older elite nodes, we include only a concise delta summary $\delta^{(1)}$. This provides a lightweight view of past strategy shifts without exposing implementation details.

---

[3]MAP-Elites (Multi-dimensional Archive of Phenotypic Elites) is a quality-diversity algorithm that partitions the solution space into cells based on behavior descriptors and retains the highest-performing solution within each cell.

- **Level 2 (Recent Insights).** For more recent nodes and all selected elite or diverse inspirations, we include the delta plan details $\delta^{(2)}$, which describe the concrete logic changes and underlying hypotheses that proved effective.
- **Level 3 (Immediate Context).** For the current parent node, we include the full executable code $p_{\text{parent}}$ to support direct and valid code modification.

This progressive disclosure mechanism preserves informative evolutionary signals while further reducing token usage, with the detailed algorithm described in Algorithm 1.

## 5. Experiments

In this section, we conduct experiments to answer the following questions:

- **RQ1 (Solution Quality):** *Can* `DeltaEvolve` *discover solutions with superior objective scores compared to state-of-the-art evolutionary baselines across diverse scientific domains?*
- **RQ2 (Token Consumption):** *Does* `DeltaEvolve` *reduce the computational cost (input tokens) required to discovery high-quality solutions?*

### 5.1. Setup

**Tasks.** We evaluate open-ended problems across diverse domains: (1) `BlackboxOptimization` (Finck et al., 2009): minimize the objective value of five standard black-box functions (e.g., Rosenbrock, Rastrigin) with input dimensions scaling from 3 to 40; (2) `HexagonPacking` (Georgiev et al., 2025): pack $N = 11$ unit regular hexagons into the smallest possible outer regular hexagon without overlap; (3) `SymbolicRegression` (Shojaee et al., 2025): discover the mathematical expression that best fits a given dataset by uncovering latent relationships; (4) `PDESolver` (Li et al., 2025): evolve numerically stable iterative linear solvers to minimize the residual norm of $Ax = b$ for large sparse systems; and (5) `EfficienctConvolution` (Press et al., 2025): minimize the wall-clock execution time of 2D convolution kernels on dynamic problem scales ($30n \times 30n$) subject to strict correctness verification.

**Baselines.** We compare `DeltaEvolve` against other code-generation paradigms: (1) `Parallel Sampling`: a Best-of-$N$ sampling strategy that selects the highest-scoring candidate from independent samples; (2) `Greedy Refine`: iterative modification of the current best solution based on verbal feedback from the evaluator; and (3) `AlphaEvolve` (Novikov et al., 2025): a representative state-of-the-art evolutionary agent that relies on full-code history; we reproduce it using the open-source implementation of `OpenEvolve` (Sharma, 2025).

*Table 2.* Comprehensive evaluation on tasks across 5 domains. We compare `DeltaEvolve` against four distinct baselines: (1) `Parallel Sampling` (Best-of-N sampling), (2) `Greedy Refine` (Refinement top code), and (3) `AlphaEvolve` (Evolutionary search with full code). `Best Score` reports the objective value. `Token Consump.` represents the number of total used tokens.

| Task Domain | Method | GPT-5-mini + o3-mini | | Gemini-2.5-flash-lite + Gemini-2.5-flash | |
|---|---|---|---|---|---|
| | | Best Score(↑) | Token Consump.(↓) | Best Score(↑) | Token Consump.(↓) |
| **1. BlackBox Optimization** | Parallel Sampling | 0.4161 | 390872 | 0.4161 | 390872 |
| | Greedy Refine | 2.2618 | 555430 | 2.3403 | 1054566 |
| | AlphaEvolve (Full Code) | 2.6415 | 1852841 | 2.5221 | 1894890 |
| | **DeltaEvolve (Ours)** | **2.7297** | 1390709 | **3.9372** | 1227388 |
| **2. Hexagon Packing** | Parallel Sampling | 0.4913 | 565016 | 0.4913 | 565016 |
| | Greedy Refine | 0.8508 | 717697 | 0.4913 | 814092 |
| | AlphaEvolve (Full Code) | 0.9721 | 905249 | 0.7859 | 1334339 |
| | **DeltaEvolve (Ours)** | **0.9821** | 827884 | **0.8804** | 893749 |
| **3. Symbolic Regression** | Parallel Sampling | 1.8527 | 503098 | 1.8535 | 503098 |
| | Greedy Refine | 2.9553 | 811404 | 2.9550 | 756762 |
| | AlphaEvolve (Full Code) | 3.2657 | 1660699 | 3.2174 | 1545091 |
| | **DeltaEvolve (Ours)** | **3.4174** | 810354 | **3.2198** | 832164 |
| **4. PDE Solver** | Parallel Sampling | 0.7506 | 154016 | 0.7506 | 154016 |
| | Greedy Refine | 0.7506 | 375186 | 0.7506 | 255332 |
| | AlphaEvolve (Full Code) | 0.8850 | 711298 | 0.9901 | 595094 |
| | **DeltaEvolve (Ours)** | **0.8915** | 562848 | **0.9931** | 253719 |
| **5. Efficient Convolution** | Parallel Sampling | 0.8035 | 139958 | 0.8036 | 139958 |
| | Greedy Refine | 0.8035 | 240473 | 0.8037 | 305334 |
| | AlphaEvolve (Full Code) | 0.8974 | 683439 | 0.8219 | 885592 |
| | **DeltaEvolve (Ours)** | **0.9067** | 348539 | **0.9032** | 517281 |

**Implementations.** Following AlphaEvolve (Novikov et al., 2025), we employ an LLM ensemble strategy with probability of 0.8 for high-throughput generation, and 0.2 for complex reasoning across two distinct model families: (1) `gpt-5-mini` paired with `o3-mini`, and (2) `gemini-2.5-flash-lite` paired with `gemini-2.5-flash`. For the `Progressive Disclosure Sampler`, we configure the prompt context to include $k = 3$ top-performing nodes and $m = 2$ diverse nodes. Detailed configurations are provided in Table 4 in the Appendix.

### 5.2. Evaluation Metrics

**1. Best Score (↑).** We report the raw objective value of the best-performing program discovered within the fixed budget, serving as the direct measure of solution quality.

**2. Token Consumption (↓).** We measure the computational cost of discovery by the cumulative number of tokens consumed throughout the search process. Token consumption is defined as the total number of tokens used across all language model calls and all iterations, including parallel samples. Lower token consumption indicates lower overall computational cost.

### 5.3. Main Results

Section 4.3 reports results across all task domains. To ensure robustness, we run each method with three random seeds $(11, 42, 100)$. We report the maximum `best score` to capture best solutions achieved and the average `token consumption` to measure cost-effectiveness. `DeltaEvolve` consistently outperforms all baselines on both metrics. Although gains in `best score` may appear small for some tasks, improvements near theoretical or best-known optima are increasingly difficult, making even modest gains meaningful. In terms of cost, `DeltaEvolve` achieves substantially lower `token consumption` than AlphaEvolve, validating our core design: replacing redundant full-code snapshots with compact `semantic delta` representations reduce token consumption, enabling faster and more directed evolution.

## 6. Conclusion

We present `DeltaEvolve`, a novel momentum-driven framework that addresses the context inefficiency and weak evolutionary signals of existing full-code self-evolving systems. By formalizing program evolution as an Expectation-Maximization process, we introduce `semantic delta` as a transferable and informative driver of improvement. Together, the multi-level database and the progressive disclosure sampler enable a more efficient use of context. Experiments demonstrate that `DeltaEvolve` consistently outperforms state-of-the-art across diverse scientific domains, achieving superior solution quality while significantly improving token efficiency. Our work highlights the critical role of momentum-based history in overcoming the context bottleneck for automated scientific discovery.

## Acknowledgements

We acknowledge support from NSF grants IIS-2312840 and IIS-2402952. We gratefully acknowledge Ismail Alkhouri, Yuxin Dong, Xia Ning, and Huan Sun for valuable discussions.

## Impact Statement

This work presents `DeltaEvolve`, a framework designed to accelerate automated scientific discovery through momentum-driven evolution. Our research does not involve the collection of new human or animal data, and all experiments are conducted using publicly available mathematical and algorithmic benchmarks. We acknowledge that the pre-trained LLMs employed as the generative backbone may inherit biases and safety risks present in their training data. Since our method optimizes the context via inference-time search rather than model fine-tuning, it does not explicitly mitigate such inherent biases. Furthermore, given the agent's capability to generate and execute code autonomously, we emphasize the necessity of secure, sandboxed execution environments to prevent unintended system behaviors. We encourage future research to examine the safety, security, and ethical implications of deploying self-evolving coding agents in real-world scientific workflows.

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

# Appendix

## A. Optimization vs. Algorithmic Discovery

**Concrete Example of Optimization and Algorithmic Discovery.** Table 3 highlights the structural mismatch between classical optimization and algorithmic discovery. While classical methods optimize continuous parameters using analytical gradients, algorithmic discovery searches over discrete program spaces using evaluator-based feedback. Improvements therefore arise from semantic changes in program logic rather than numerical updates, motivating evolution signals that explicitly capture how programs change and why they improve.

*Table 3.* Concrete form comparison: Classical Optimization vs. Algorithmic Discovery.

|  | Classical Optimization
*(e.g., Least Squares)* | Algorithmic Discovery
*(e.g., Circle Packing)* |
|---|---|---|
| (i) Objective | $\min_{x} \|Ax - b\|_2^2$ | `max sum(radius of circles)`
`s.t.  no overlap` |
| (ii) Space | $x = [x_1, \ldots, x_n]^\top$ | `def pack(box, n):  ...  return` |
| (iii) Feedback | $\nabla f = 2A^\top (Ax - b)$
*(Gradient Vector)* | `Score:  0.65`
*(Evaluator Information)* |
| (iv) Solution | $x^* = [0.01, -1.2]^\top$ | `if box.w < 10:  spiral()`
`else:  greedy_place()` |

## B. Algorithm of Progressive Disclosure Sampler

Algorithm 1 presents the progressive disclosure sampler that adaptively renders historical nodes at different abstraction levels to retain key evolution signals while minimizing context length.

---

**Algorithm 1** Progressive Disclosure Sampler

---

**Require:** Database $\mathcal{D} = \{N_1, \ldots, N_T\}$, where $N_t = (\delta_t^{(1)}, \delta_t^{(2)}, p_t)$;
    Parent selection policy $\pi_{\text{select}}$;
    Maximum elites $k$, maximum inspirations $m$;
    Recent window size $w$
**Ensure:** Final prompt context $C_{\text{final}}$

1: ______________________________________________
2:   *// Phase 1: Node Selection*
3:   $N_{\text{parent}} \leftarrow \text{Sample}(\mathcal{D}, \pi_{\text{select}})$
4:   $\mathcal{N}_{\text{elite}} \leftarrow \text{TopK}(\mathcal{D}, R(p), k)$
5:   $\mathcal{N}_{\text{div}} \leftarrow \text{MAPElitesSample}(\mathcal{D}, \text{exclude} = N_{\text{parent}}, m)$
6:
7:   *// Phase 2: Progressive Rendering*
8:   $C_{\text{hist}} \leftarrow \emptyset$
9:   **for** $N_i \in (\mathcal{N}_{\text{elite}} \cup \mathcal{N}_{\text{div}})$ **do**
10:     **if** $\text{Time}(N_i) > t - w$ **then**
11:       $C_{\text{hist}} \leftarrow C_{\text{hist}} \oplus \text{Format}(\delta_i^{(2)})$          *// Level 2: delta plan*
12:     **else**
13:       $C_{\text{hist}} \leftarrow C_{\text{hist}} \oplus \text{Format}(\delta_i^{(1)})$          *// Level 1: summary*
14:     **end if**
15:   **end for**
16:   $C_{\text{final}} \leftarrow C_{\text{hist}} \oplus \text{Format}(p_{\text{parent}})$          *// Level 3: full code*
17:
18:   **return** $C_{\text{final}}$

---

# C. Prompt Templates

This appendix section provides the exact system and user prompts used by the `DeltaEvolve` pipeline, highlighting the key architectural shifts from standard baselines like AlphaEvolve. In the **System Prompt**, we introduce a novel "Delta Logging Instructions" component; this explicitly constraints the LLM to not only generate code but also synthesize a structured `DELTA-SUMMARY` (Level 1) and `DELTA-PLAN` (Level 2) post-modification, effectively enabling the agent to self-document its evolutionary logic.

---

**Symtem Prompt Template**

```
------------------------------- Problem Description -------------------------------
{problem_details}

--------------------- Delta Logging Instructions (New in DeltaEvolve) ---------------
You MUST summarize your changes at the very end of your response using the strict
format below.  This log is critical for the evolution memory system.  Failure to
follow these rules will break the experiment.

###CRITICAL RULES for DELTA SUMMARY:
1.  NO META-TALK: Do NOT say Replace 22 lines with 230 lines, Updated code, or Changed
logic.
2.  ALGORITHMIC ONLY: Describe the strategy change (e.g., Switched from Greedy to
Simulated Annealing).
3.  NO TEMPLATES: Do NOT output placeholder text like high-level plan summary in one
sentence.  Write actual content.
4.  NO CODE SYNTAX: Do not write Python code in the summary (e.g., no def function()).
Use natural language.

#DELTA-SUMMARY-START
FROM: <One-sentence summary of the OLD strategy (the parent node's approach)>
TO: <One-sentence summary of the NEW strategy (your current approach)>
#DELTA-SUMMARY-END
-------------------------------------------------------------------------------
###CRITICAL RULES for DELTA PLAN DETAILS
You must explain HOW and WHY the logic changed using the strict format below.
1.  Target Audience:  A researcher trying to reproduce your experiment without seeing
the code.
2.  BE QUANTITATIVE: Do not say increased parameters.  Say increased grid_resolution
from 10 to 50.
3.  NAME THE ALGORITHM: Use standard terminology (e.g., Coordinate Descent, Simulated
Annealing, Penalty Method).
4.  NO META-TALK: Do not say I defined a function.  Describe the LOGIC flow.

#DELTA-PLAN-DETAILS-START
[Modification 1]
COMPONENT: <The specific module changed.  e.g., Initialization Strategy, Constraint
Handling, Optimization Loop>
OLD_LOGIC: <Brief summary of what was removed.  e.g., Random noise injection>
NEW_LOGIC: <DETAILED mechanism.  MUST include specific hyperparameters, formulas used,
or heuristic rules.>
HYPOTHESIS: <The scientific reasoning.>

[Modification 2] (If applicable)
COMPONENT: ...
OLD_LOGIC: ...
NEW_LOGIC: ...
HYPOTHESIS: ...
#DELTA-PLAN-DETAILS-END
```

*Figure 5.* **System Prompt Template.** Includes novel instructions requiring the model to output structured Delta Summaries (Level 1) and Delta Plans (Level 2) alongside code, enabling automated evolutionary logging.

The **User Prompt** is restructured to replace the token-heavy full source code of inspiration programs with a "Delta of History Nodes" section. Instead of raw implementation details, this section feeds the agent a concise trajectory of past successful strategies and rationales, maximizing context efficiency while providing a clear directional signal for future mutations.

---

**User Prompt Template**

```
-------------------------- Current Program Information --------------------------
- Focus areas: {improvement_areas}
- Feedback: {evaluator_feedback}
------------ Inspirations: Delta of History Nodes (New in DeltaEvolve) ------------
Below we provide history delta plans from prior nodes, including top-performing
programs and highly diverse alternatives to the parent. Each delta highlights
concrete strategy differences that may inspire new solution directions rather than
direct reuse.

### Top Delta Plan 1
{Delta summaries or Delta plans.}
- Feedback: {evaluator_feedback}

### Top Delta Plan 2
{Delta summaries or Delta plans.}
- Feedback: {evaluator_feedback}

### Top Delta Plan 3
{Delta summaries or Delta plans.}
- Feedback: {evaluator_feedback}

### Diverse Delta Plan 1
{Delta summaries or Delta plans.}
- Feedback: {evaluator_feedback}

### Diverse Delta Plan 2
{Delta summaries or Delta plans.}
- Feedback: {evaluator_feedback}

-------------------------------- Parent Program --------------------------------
{parent_program}

Suggest improvements to the program that will improve its FITNESS SCORE. The system
maintains diversity across these dimensions: complexity, diversity Different
solutions with similar fitness but different features are valuable.
You MUST use the exact SEARCH/REPLACE diff format shown below to indicate changes:
<<<<<<< SEARCH
<Original code to find and replace (must match exactly)>
=======
<New replacement code>
>>>>>>> REPLACE
```

---

*Figure 6.* **User Prompt Template.** Replaces full-code inspiration contexts with concise Delta Summaries and Plans, exposing evolutionary trajectories while minimizing token usage.

# D. Task Details

## D.1. Blackbox Optimization

**Task Description** The Blackbox Optimization task involves minimizing continuous objective functions drawn from the BBOB (Black-Box Optimization Benchmarking) suite, which encompasses a diverse range of landscapes including separable, ill-conditioned, and multi-modal functions. The optimizer must operate as a true black box, receiving only the problem dimension, bounds, and evaluation budget without access to gradients or the analytical form of the function. The objective is to locate the global optimum with high precision and efficiency across various function identifiers and instance realizations, while strictly adhering to search space bounds and ensuring reproducibility via fixed random seeds.

**Initial Programs** The initial program implements a baseline Random Search algorithm that explores the solution space by sampling candidate points uniformly within the hyperbox defined by the problem bounds(range of $[-5, 5]$). It dynamically detects the problem dimensionality and employs a simple "keep-best" logic, where the solution with the lowest observed function value is retained. This approach is purely stochastic and memory-less, serving as a fundamental baseline that does not adapt any search strategy.

**Evaluator** We evaluated the optimizer on five BBOB functions with increasing complexity: `sphere_d3_i1`, `rosenbrock_d5_i2`, `rastrigin_d10_i5`, `ellipsoid_d20_i1`, and `schaffers_d40_i5`. The identifiers follow the format `<function name>_<input dimension>_<instance ID>`. Reference values ($V_{\text{ref}}$) were computed using `scipy.optimize`. The evaluation employs a two-stage protocol: Stage 1 acts as a validity filter, allowing only successful runs to proceed to Stage 2 for final scoring. The final score combines solution quality and efficiency:

$$S_{\text{case}} = 0.7 \cdot S_{\text{val}} + 0.3 \cdot \max\left(0, 1 - \frac{N_{\text{used}}}{N_{\text{budget}}}\right) \qquad (11)$$

The value score $S_{\text{val}}$ depends on the normalized improvement $\delta = (V_{\text{ref}} - V_{\text{best}})/|V_{\text{ref}}|$, calculated as $1 + \delta$ if $V_{\text{best}} \leq V_{\text{ref}}$ (better or equal), and $(1 + |\delta|)^{-1}$ otherwise.

**Evolution Process** To clearly illustrate the evolutionary dynamics, we select a representative run using a fixed random seed (42). Figure 7 visualizes the optimization process, comparing the score of the combined score (left) and the cumulative token consumption (right) across iterations. The results demonstrate that `DeltaEvolve` achieves a significantly higher final score while maintaining lower token usage compared to AlphaEvolve, highlighting its superior efficiency in exploring the code space.

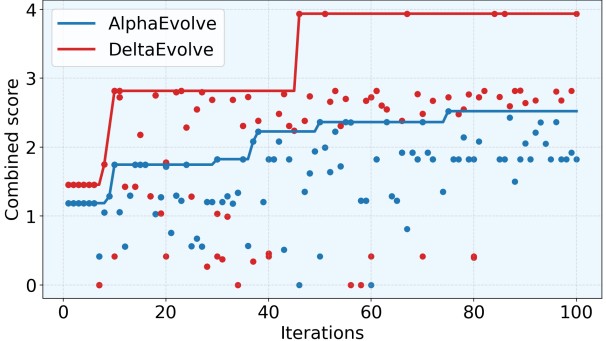 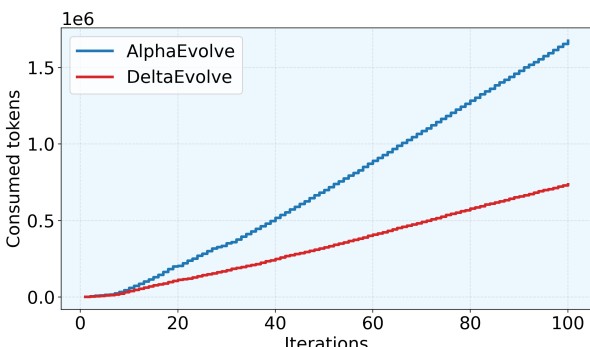

*Figure 7.* Evolutionary trajectory for a representative run. The left panel shows the score progression, and the right panel tracks token usage per iteration. `DeltaEvolve` reaches higher performance with reduced token cost.

## D.2. Hexagon Packing

**Task Description** The Hexagon Packing task requires constructing an optimal arrangement of $N = 11$ disjoint unit regular hexagons (side length $s = 1$) within a minimal bounding regular hexagon. The objective is to maximize the inverse of the outer hexagon's side length, $\rho = 1/R$, which is equivalent to maximizing the packing density. The optimizer outputs

the coordinates $(x_i, y_i)$ and rotation angles $\theta_i$ for all $N$ inner hexagons, as well as the geometry of the outer hexagon. A valid solution must satisfy strict geometric constraints: all pairs of inner hexagons must be disjoint, and all inner hexagons must be fully contained within the outer boundary.

**Initial Programs**    The initial program implements a deterministic, heuristic approach. It arranges the 11 unit hexagons in a static, pre-calculated grid pattern (a "lattice" layout) centered at the origin. The outer hexagon side length $R$ is set to a conservative value ($R = 8$) to guarantee containment. While valid, this solution is sparse and far from optimal, serving as a lower-bound baseline for the packing density.

**Evaluator**    The evaluator validates the geometric feasibility of the proposed packing using the Separating Axis Theorem (SAT). It performs $O(N^2)$ pairwise checks to ensure disjointness and $O(N)$ checks for containment within the outer boundary. If any constraint is violated (overlap or protrusion) within a numerical tolerance of $10^{-6}$, the solution is marked as invalid (score 0). Valid solutions are scored based on the inverse radius $\rho = 1/R$, normalized against a state-of-the-art benchmark ($\rho_{\text{ref}} \approx 0.2544$). The final metric balances the packing density and the computational time required to generate the solution.

**Evolution Process**    To clearly illustrate the evolutionary dynamics, we select a representative run using a fixed random seed (42). Figure 8 displays the progression of the packing density score (left) and token consumption (right). `DeltaEvolve` rapidly moves beyond the initial lattice configuration, employing a continuous relaxation strategy that iteratively resolves overlaps while compressing the outer boundary. This allows it to approach the theoretical packing limit more closely and efficiently than the baseline random search.

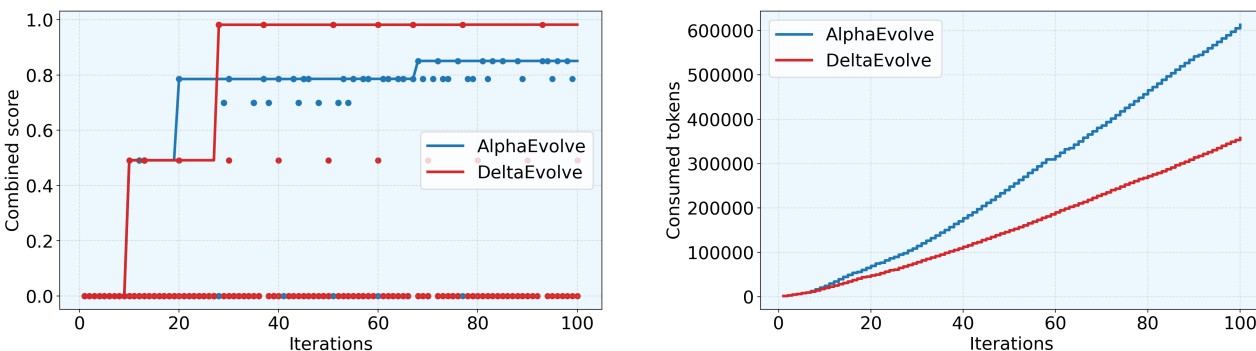

*Figure 8.* Optimization trajectory for the Hexagon Packing task. The left panel illustrates the improvement, while the right panel shows the token efficiency of the search process.

### D.3. Symbolic Regression

**Task Description**    The Symbolic Regression task involves discovering a mathematical expression that accurately models a physical process based on observational data. Specifically, the objective is to predict the acceleration ($dv/dt$) of a nonlinear harmonic oscillator given input features: position ($x$), time ($t$), and velocity ($v$). The optimizer must evolve the functional form of a Python function `func(x, params)`, where `x` is the input matrix and `params` is a vector of learnable coefficients (up to 10). This constitutes a hybrid optimization problem: the evolutionary algorithm searches for the optimal symbolic structure (e.g., polynomial terms, trigonometric functions), while a numerical optimizer adjusts the coefficients to fit the data.

**Initial Programs**    The initial program implements a naive linear model: $f(\mathbf{x}, \mathbf{p}) = p_0 \cdot x + p_1 \cdot t + p_2 \cdot v$. While computationally efficient and easy to optimize, this model is fundamentally insufficient for capturing the dynamics of a nonlinear oscillator, which typically requires restoring forces (dependent on position) and damping terms (dependent on velocity) that interact non-linearly (e.g., cubic stiffness $x^3$ in a Duffing oscillator).

**Evaluator**    The evaluation pipeline measures the predictive accuracy of the proposed model on a training dataset. For every candidate functional structure, the evaluator employs the BFGS algorithm (`scipy.optimize.minimize`) to

optimize the vector `params` by minimizing the Mean Squared Error (MSE) between the predicted acceleration and the ground truth. The model's fitness is quantified as $-\log_{10}(\text{MSE})$, promoting high-precision solutions. The evaluator also enforces robustness checks, assigning penalizing scores to models that generate numerical errors (e.g., `NaN`, overflow) or fail to execute within a strictly defined timeout.

**Evolution Process**   We analyze the search trajectory using a fixed random seed. Figure 9 depicts the improvement in the model's score (left) and the token consumption (right) over iterations. `DeltaEvolve` successfully transitions from the linear baseline to a non-linear formulation—identifying critical terms like cubic stiffness or interaction effects—resulting in a sharp reduction in MSE and a corresponding increase in the logarithmic score, all while maintaining concise code expressions.

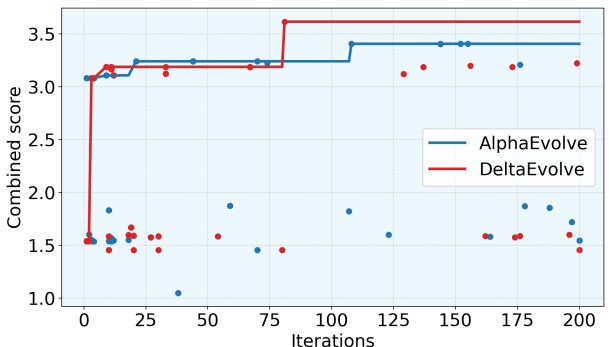 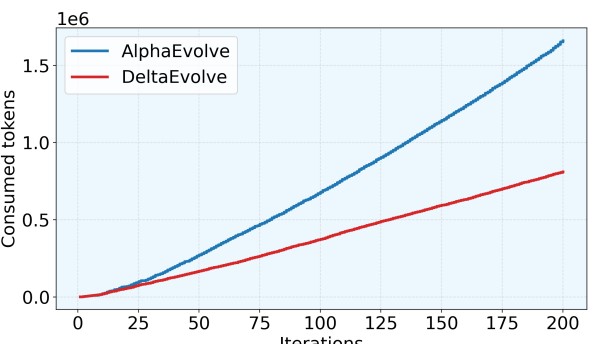

*Figure 9.* Evolutionary progress for Symbolic Regression. The left panel shows the score, and the right panel tracks token usage. The optimizer quickly identifies the necessary nonlinear terms to fit the physical dynamics.

### D.4. PDE Solver

**Task Description**   The objective is to optimize a Krylov subspace solver for sparse linear systems of the form $Ax = b$. The specific focus is on the Conjugate Gradient (CG) method, which is the standard solver for Symmetric Positive Definite (SPD) systems (e.g., discretized Poisson equations). The optimizer must enhance the solver to reduce the number of iterations required for convergence and improve numerical stability against ill-conditioned matrices.

**Initial Programs**   The initial program implements a standard Conjugate Gradient algorithm with a basic Jacobi (diagonal) preconditioner. It computes the inverse diagonal $M^{-1} = \text{diag}(A)^{-1}$ to scale the residual. While computationally cheap, the Jacobi preconditioner provides only modest improvements in convergence for coupled systems like the Poisson equation. The baseline implementation lacks restart mechanisms, making it susceptible to loss of orthogonality in high-dimensional or ill-conditioned problems.

**Evaluator**   The evaluator assesses the solver on the 2D Poisson equation discretized on square grids of sizes $N \in \{50, 100, 200\}$. This generates sparse SPD matrices with condition numbers scaling as $O(N^2)$. Performance is scored based on a composite metric of convergence speed (number of iterations), solution accuracy ($L_2$ error relative to ground truth), and the final residual norm. The score penalizes slow convergence exponentially: $S \propto e^{-k/N}$, encouraging solvers that drastically reduce the iteration count $k$.

**Evolution Process**   We visualize the evolution using a fixed random seed. Figure 10 compares the combined score (left) and token usage (right). `DeltaEvolve` rapidly identifies that the baseline Jacobi preconditioner is the bottleneck. It evolves a more sophisticated Polynomial Preconditioner (approximating $A^{-1}$ via a Neumann series) and introduces a residual-based restart heuristic. This results in a solver that converges in significantly fewer iterations than the baseline, achieving a higher score with efficient code usage.

### D.5. Efficient Convolution.

**Task Description**   This task focuses strictly on computational efficiency. It involves optimizing a 2D convolution operation, a fundamental kernel in signal processing and computer vision. The inputs are two real-valued matrices: a large input array

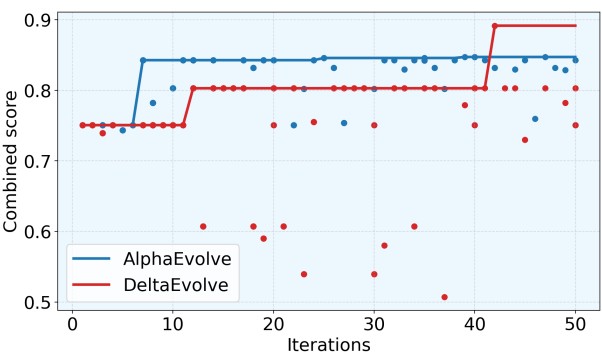 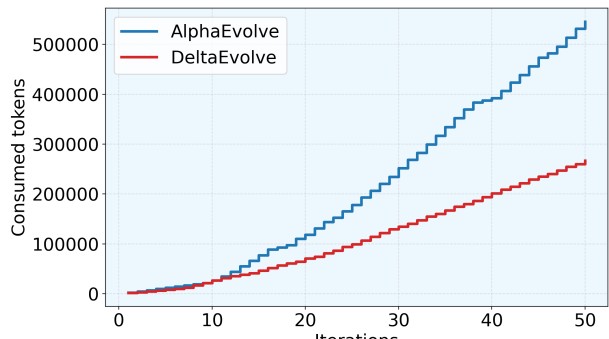

*Figure 10.* Optimization trajectory for the PDE Solver. The left panel tracks the combined accuracy/speed score, and the right panel tracks token usage. The `DeltaEvolve` achieves better combined score and less token usage.

$A$ of size $(30n \times 30n)$ and a smaller kernel $B$ of size $(8n \times 8n)$, where $n$ is a scaling factor. The operation is defined in "full" mode with zero-padding ("fill" boundary handling), meaning the output size is $(H_A + H_B - 1) \times (W_A + W_B - 1)$. The objective is to maximize the speedup relative to a reference `scipy.signal.convolve2d` implementation while maintaining numerical correctness ($L_2$ relative error $< 10^{-6}$).

**Initial Programs**  The initial program is a direct wrapper around `scipy.signal.convolve2d`, providing a functional baseline but lacking any specific optimizations for the problem structure. It operates in the spatial domain, which scales as $O(N_A^2 N_B^2)$, becoming computationally prohibitive for large $n$. The baseline does not leverage parallelization, vectorization, or frequency-domain transformations that could significantly accelerate the computation.

**Evaluator**  The evaluator benchmarks the evolved solution against the reference `AlgoTune` baseline. The performance metric is the `speedup ratio`: Speedup $= T_{\text{baseline}}/T_{\text{evolved}}$. Correctness is verified by comparing the Frobenius norm of the difference between the evolved output and the reference output. The evaluation pipeline includes warmup runs to stabilize JIT compilers and multiple timing trials to ensure robust measurements. Solutions that exceed a timeout or fail the accuracy check receive a zero score.

**Evolution Process**  We analyze the optimization trajectory using a fixed random seed. Figure 11 illustrates the speedup achieved (left) and the token consumption (right). `DeltaEvolve` successfully transitions from the spatial domain baseline to a frequency-domain approach using the Fast Fourier Transform (FFT). By exploiting the Convolution Theorem, it reduces the complexity to $O(N^2 \log N)$, yielding a dramatic speedup for large inputs. It further optimizes memory layout and FFT planning to maximize efficiency.

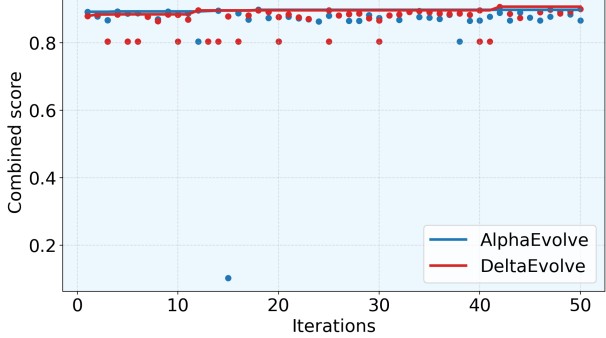 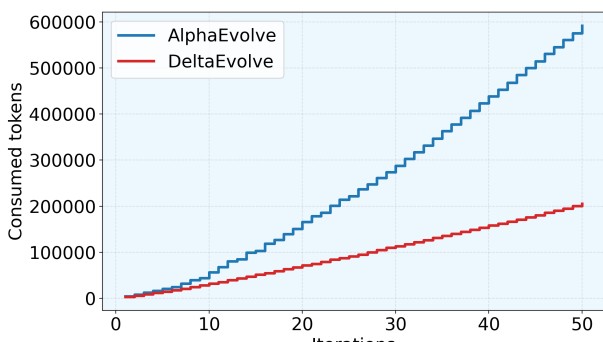

*Figure 11.* Optimization trajectory for the Convolve2D task. The left panel tracks the efficiency score (speedup), while the right panel tracks token usage. `DeltaEvolve` identifies the optimal FFT approach, significantly outperforming the baseline.

# E. Case Study

To demonstrate `DeltaEvolve` in practice, we present a case study on a black-box optimization task that visualizes the full evolutionary trajectory across generations. Figure 12 plots the best-so-far combined score (higher is better) against the number of iterations, annotating key milestones with their corresponding Level 1 Delta Summaries. The trajectory reveals a coherent evolutionary logic: initially, the agent explores basic heuristics like local Gaussian perturbations (Iter 1–2). A critical structural breakthrough occurs at Iteration 14, where the agent explicitly proposes a semantic shift to "Latin Hypercube initialization with adaptive batched local search," triggering a steep performance gain ($1.68 \rightarrow 2.11$). Subsequently, rather than randomly rewriting the codebase, the agent leverages the delta history to perform targeted refinements—introducing "stagnation probing" (Iter 41) and "Metropolis acceptance" (Iter 51) to optimize budget usage and escape local optima. This progression confirms that `DeltaEvolve` actively maintains evolutionary momentum, iteratively stacking meaningful logical improvements to converge on a robust final solution.

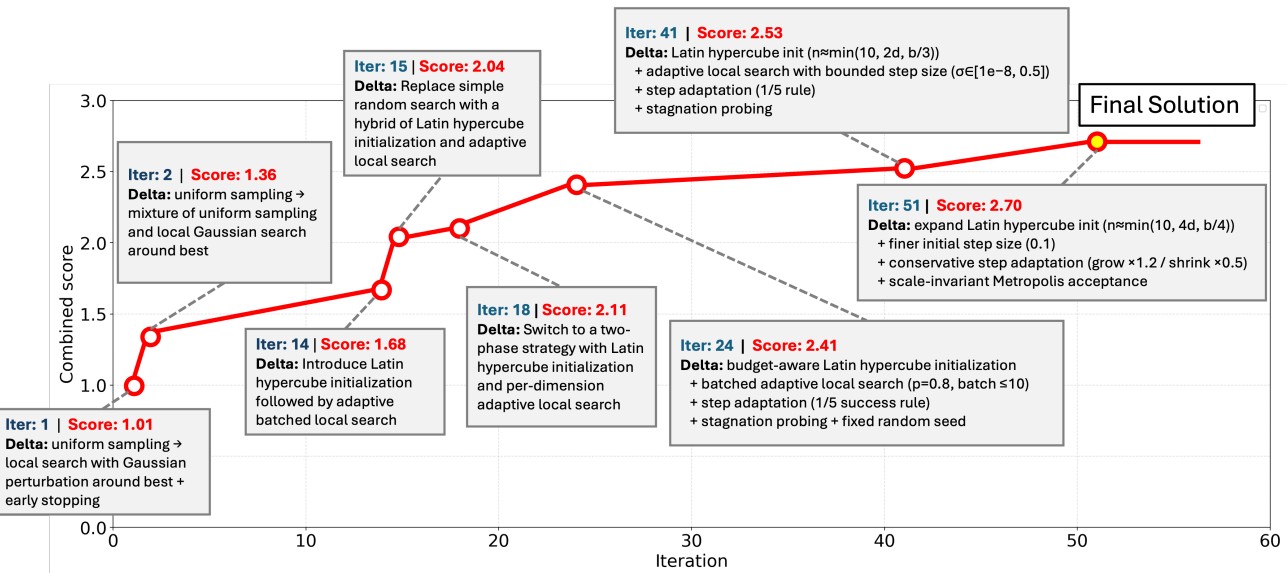

*Figure 12.* **Case Study of DeltaEvolve Dynamics on Black-Box Optimization.** We plot the best-so-far combined score (higher is better) versus iterations. Annotated nodes show the **Delta Summary (Level 1)** between successive solutions, illustrating how changes in initialization, adaptive local search, and acceptance mechanisms are progressively introduced.

## F. Examples of Semantic Deltas

To provide concrete intuition for our multi-level representation, we present real-world artifacts generated by `DeltaEvolve` during the Black-Box Optimization task. Figure 13 displays a **Level 1 Delta Summary**, which captures the high-level strategic pivot in a concise `FROM/TO` format—summarizing the transition from random sampling to a two-phase budget-aware optimizer without unnecessary detail. Complementing this, Figure 14 shows the corresponding **Level 2 Delta Plan Details**, which acts as a structured scientific log. It breaks down the evolution into specific components (e.g., Initialization, Step-size), explicitly contrasting `OLD_LOGIC` with `NEW_LOGIC` and providing the underlying hypothesis. These readable logs allow the agent to perceive the causal mechanism of improvement without parsing the full source code.

---

**Delta Summary (Level 1)**

```
[iter:  027 | improved]
FROM: Uniform random sampling with occasional fixed-size local perturbations (5
Gaussian tries at 5% range) and greedy replacement.

TO: A two-phase budget-aware optimizer:  Latin Hypercube initialization (n_init =
min(max(10,4*dim), max(1,budget//3))) followed by an adaptive batched per-dimension
Gaussian local search (sigma init = 0.2*span, grow * 1.2 on success, shrink *0.5 on
stagnation with patience = max(3,dim), batch = min(10, remaining//10), exploitation
p=0.8) plus occasional global probes and strict budget accounting.

Evaluation Results:{ stage1_cases:  { "sphere_d3_i1":  0.699 } | stage2_cases:  {
"rosenbrock_d5_i2":  0.703, "rastrigin_d10_i5":  4.354, "ellipsoid_d20_i1":  1.270,
"schaffers_d40_i5":  0.700 } }
```

---

*Figure 13.* **Example of Delta Summary (Level 1).** Generated during Black-Box Optimization, this summary captures the high-level strategic pivot (FROM random sampling TO adaptive LHS) in a concise, token-efficient format suitable for long-term history retrieval.

---

**Delta Plan Details (Level 2)**

```
## Program Top 1:
[iter:038:improved]
[Delta plan details]:

[Modification 1]
COMPONENT: Initialization Strategy (Latin Hypercube)
OLD_LOGIC: n_init = min(max(10, 4*dimension), max(1, budget//4)); per-dimension shuffle
of midpoints with jitter = (rng.random - 0.5)/n_init.
NEW_LOGIC: n_init = min(max(10, 4*dimension), max(1, budget//3)), then clamp
n_init < max(1, budget-2) to leave two evaluations.  Construct full LHS by
computing base midpoints base[i] = (i+0.5)/n_init, generating a random permutation
idx per-dimension, forming perms[:,d] = base[idx], then adding jitter =
(rng.random((n_init,dimension))-0.5)/(n_init*3.0) and clipping to [0,1].  Map to box
via lower + sample*span.
HYPOTHESIS: Full-permutation LHS with slightly more initialization samples improves
space-filling coverage; smaller jitter reduces boundary bias while keeping diversity;
leaving two evals ensures immediate subsequent polish/search.

[Modification 2]
COMPONENT: Initialization Polishing (Coordinate Refinement)
OLD_LOGIC: Single small-radius sweep:  refine_evals = min(budget-evals, min(2*dim,10)),
radius = 0.05*span, try +/- on each coordinate once.
NEW_LOGIC: Multi-scale polish with refine_evals = min(budget-evals, min(3*dim,15)).
Iterate scales = [0.2, 0.05, 0.01]; for each scale, compute radius = scale*span and
for each dimension try +/- radius while budget remains; accept and update best, and
bias current_x/current_val to improvements immediately.
HYPOTHESIS: Many BBOB functions are separable or have strong one-dimensional curvature;
coarse-to-fine coordinate probes (0.2 then 0.05 then 0.01) cheaply discover large
improvements early and avoid wasting many evaluations at a single tiny scale.
```

```
[Modification 3]
COMPONENT: Step-size Initialization and Lower Bound
OLD_LOGIC: initial_step = 0.1*span; min_step = 1e-8*span; grow=1.3/shrink=0.7.
NEW_LOGIC: initial_step = 0.2*span; min_step = 1e-6*span (keep grow=1.3, shrink=0.7);
clamp step updates as before but with the new numeric bounds.
HYPOTHESIS: A slightly larger initial step accelerates early descent and exploration;
enforcing a minimum of 1e-6*span reduces chance of adaptation getting stuck on
numerically negligible steps, improving robustness across varying dimensions.

[Evaluation results] {"sphere_d3_i1":  0.699, "rosenbrock_d5_i2":  0.704,
"rastrigin_d10_i5":  6.024, "ellipsoid_d20_i1":  1.343, "schaffers_d40_i5":  0.709}
```

*Figure 14.* **Example of Delta Plan (Level 2).** This structured log details specific algorithmic modifications (Initialization, Polishing, Step-size), explicitly contrasting OLD_LOGIC versus NEW_LOGIC to expose the causal mechanism of improvement to the LLM.

## G. Configuration of DeltaEvolve

**Default Configuration.**  Unless otherwise stated, all experiments use the default configuration summarized in Table 4.

*Table 4.* System configuration and hyperparameters.

| Parameter | Value | Parameter | Value |
|---|---|---|---|
| **LLM settings** | | | |
| Primary model | gpt-5-mini | Secondary model | o3-mini |
| Temperature | 0.7 | Top-p | 0.95 |
| Max tokens | 8192 | Timeout | 600s |
| **Prompt** | | | |
| Number of top plans | 3 | Number of Diverse Plans | 2 |
| Parent selection strategy | weighted | Elite selection ratio | 0.1 |
| Exploration ratio | 0.2 | Exploitation ratio | 0.7 |
| **Database** | | | |
| Population size | 40 | Archive size | 20 |
| Top-$k$ inspirations | 3 | Diverse inspirations | 2 |
| Migration interval | 10 | Migration Rate | 0.1 |
| Number of islands | 3 | Parent selection strategy | weighted |
| **Evolution** | | | |
| Max Iterations | 100 | Random seed | 42 |
| **Evaluator** | | | |
| Timeout | 300s | Max retries | 3 |
| Parallel evaluations | 4 | Cascade evaluation | False |

# H. Discovered Solutions

For blackbox optimization, `DeltaEvolve` discovery a new solution that achieves a score of 3.937, significantly surpassing the AlphaEvolve baseline of 2.6415. This gain stems from implementing a robust CMA-ES (Covariance Matrix Adaptation Evolution Strategy), which adapts the search distribution's geometry to match the problem's underlying landscape. By learning variable correlations and dynamically adjusting step sizes, the optimizer navigates ill-conditioned and non-separable functions far more efficiently than standard heuristics. Key enhancements include budget-aware hyperparameter scaling and a sophisticated boundary handling mechanism that preserves search direction, ensuring rapid convergence even under tight evaluation constraints.

```
1   # EVOLVE-BLOCK-START
2   """Baseline black-box optimizer for BBOB problems."""
3
4   from typing import Sequence, Tuple, Union
5   import numpy as np
6
7
8   def _get_bounds(problem, dimension: int) -> Tuple[np.ndarray, np.ndarray]:
9       """Extract bounds from the problem or fall back to a symmetric box."""
10      lower = getattr(problem, "lower_bounds", None)
11      upper = getattr(problem, "upper_bounds", None)
12
13      if lower is None or upper is None:
14          lower = [-5.0] * dimension
15          upper = [5.0] * dimension
16
17      lower_arr = np.asarray(lower, dtype=float)
18      upper_arr = np.asarray(upper, dtype=float)
19
20      if lower_arr.shape[0] != dimension or upper_arr.shape[0] != dimension:
21          lower_arr = np.full(dimension, float(lower_arr.flat[0]))
22          upper_arr = np.full(dimension, float(upper_arr.flat[0]))
```

```python
23
24          return lower_arr, upper_arr
25
26
27  def _sample_uniform(rng: np.random.Generator, lower: np.ndarray, upper: np.ndarray) -> np.
        ndarray:
28      """Uniform sample inside the box."""
29      return lower + rng.random(size=lower.shape[0]) * (upper - lower)
30
31
32  def _clip_to_bounds(x: np.ndarray, lower: np.ndarray, upper: np.ndarray) -> np.ndarray:
33      return np.clip(x, lower, upper)
34
35
36  def _to_params(x: np.ndarray) -> dict:
37      """Convert vector to the dict format expected by bbob Problem.evaluate."""
38      return {f"x{i}": float(v) for i, v in enumerate(x)}
39
40
41  def _evaluate_safe(problem, x: np.ndarray) -> float:
42      """Evaluate the problem and guard against failures."""
43      try:
44          params = _to_params(x)
45          value = problem.evaluate(params)
46          value = float(value)
47          if np.isnan(value) or np.isinf(value):
48              return float("inf")
49          return value
50      except Exception:
51          return float("inf")
52
53
54  def run_search(problem, budget: int = 1000, seed: int | None = None) -> Tuple[list[float],
         float, int]:
55      """
56      CMA-ES (Covariance Matrix Adaptation Evolution Strategy) optimizer.
57      Adapts search distribution to efficiently find optimal solutions.
58      Deterministic under 'seed'.
59      """
60      rng = np.random.default_rng(seed)
61
62      dimension = getattr(problem, "dimension", None)
63      if dimension is None:
64          lower_attr = getattr(problem, "lower_bounds", [])
65          upper_attr = getattr(problem, "upper_bounds", [])
66          dimension = len(lower_attr) or len(upper_attr) or 2
67
68      lower, upper = _get_bounds(problem, dimension)
69
70      # CMA-ES parameters
71      # Initial sigma: A fraction of the search space width.
72      # Adapt this based on budget and search space characteristics.
73      average_range = np.mean((upper - lower) / 2.0)
74      min_dim_range = np.min(upper - lower) # Added for a minimum step size
75
76      if budget < 100 * dimension: # Heuristic for smaller budgets
77          initial_sigma = average_range * 0.15 # Reduced multiplier for more controlled
                exploration
78      else:
79          initial_sigma = average_range * 0.4 # Reduced multiplier for more controlled
                exploration
80
81      # Ensure a minimum initial sigma to prevent premature convergence on very small ranges
82      initial_sigma = max(initial_sigma, min_dim_range * 0.05) # Reduced floor based on min
            dimension range
```

```python
83
84       # Population size (lambda). A common choice is 4 + floor(3 * log(dimension)).
85       # Ensure population_size is at least 2 and allows for sufficient generations.
86       base_population = int(4 + np.floor(3 * np.log(dimension)))
87       population_size = max(2, base_population)
88       if budget >= 10: # Only apply budget cap if budget is reasonable
89           population_size = min(population_size, budget // 5) # More conservative cap to
                 allow more generations
90       if dimension > 20: # Cap for very high dimensions to manage computational cost
91           population_size = min(population_size, 2 * dimension)
92
93       # Initial mean: center of the search space
94       mean = (lower + upper) / 2.0
95
96       # Initialize CMA-ES state variables
97       # C: Covariance matrix, initially identity matrix
98       C = np.eye(dimension)
99       # pc: Evolution path for C
100      pc = np.zeros(dimension)
101      # ps: Evolution path for sigma
102      ps = np.zeros(dimension)
103      # B: Eigenvectors of C (for coordinate system transformation)
104      B = np.eye(dimension)
105      # D: Square root of eigenvalues of C (scaling factors)
106      D = np.ones(dimension)
107
108      # Learning rates and weights
109      # Number of parents (mu)
110      mu = population_size // 2
111      # Weights for recombination
112      weights = np.log(mu + 0.5) - np.log(np.arange(1, mu + 1))
113      weights = weights / np.sum(weights)
114      # Variance effective selection mass
115      mu_eff = np.sum(weights)**2 / np.sum(weights**2)
116
117      # Adapt learning rates (constants from CMA-ES literature)
118      # c_sigma: Learning rate for sigma
119      c_sigma = (mu_eff + 2) / (dimension + mu_eff + 5)
120      # d_sigma: Damping for sigma. Increased base value for slower step size reduction.
121      d_sigma = 2.5 + 2 * max(0, np.sqrt((mu_eff - 1) / (dimension + 1)) - 1) + c_sigma
122      # c_c: Learning rate for pc
123      c_c = (4 + mu_eff / dimension) / (dimension + 4 + 2 * mu_eff / dimension)
124      # c_1: Learning rate for C (rank-one update)
125      # Learning rates (constants from CMA-ES literature)
126      # c_sigma: Learning rate for sigma
127      c_sigma = (mu_eff + 2) / (dimension + mu_eff + 5)
128      # d_sigma: Damping for sigma. Increased base value for slower step size reduction.
129      d_sigma = 2.5 + 2 * max(0, np.sqrt((mu_eff - 1) / (dimension + 1)) - 1) + c_sigma
130      # c_c: Learning rate for pc
131      c_c = (4 + mu_eff / dimension) / (dimension + 4 + 2 * mu_eff / dimension)
132      # c_1: Learning rate for C (rank-one update)
133      c_1 = 2 / ((dimension + 1.5)**2 + mu_eff)
134      # c_mu: Learning rate for C (rank-mu update)
135      c_mu = min(1 - c_1 - c_1 / mu, 2 * (mu_eff - 2 + 1 / mu_eff) / ((dimension + 2)**2 +
             mu_eff))
136
137      evaluations_used = 0
138      best_x = None
139      best_value = float("inf")
140
141      # The main CMA-ES loop
142      while evaluations_used < budget:
143          # Generate population
144          population = []
145          for _ in range(population_size):
```

```
146              # Sample z from N(0, I)
147              z = rng.normal(size=dimension)
148              # Transform z to y in the search space
149              y = B @ (D * z)
150              # Add to mean and scale by sigma
151              candidate_raw = mean + initial_sigma * y
152
153              # Project candidates to bounds instead of clipping
154              candidate = candidate_raw
155              for i in range(dimension):
156                  if candidate[i] < lower[i]:
157                      # Project towards mean along this dimension
158                      candidate[i] = mean[i] + (lower[i] - mean[i]) * ((candidate_raw[i] -
                          mean[i]) / (lower[i] - mean[i]))
159                  elif candidate[i] > upper[i]:
160                      # Project towards mean along this dimension
161                      candidate[i] = mean[i] + (upper[i] - mean[i]) * ((candidate_raw[i] -
                          mean[i]) / (upper[i] - mean[i]))
162
163              # Fallback clip to ensure strict bounds if projection logic is imperfect
164              candidate = _clip_to_bounds(candidate, lower, upper)
165              population.append(candidate)
166
167          # Evaluate population
168          fitnesses = []
169          for candidate in population:
170              if evaluations_used >= budget:
171                  break
172              value = _evaluate_safe(problem, candidate)
173              evaluations_used += 1
174              fitnesses.append(value)
175
176              if value < best_value:
177                  best_value = value
178                  best_x = candidate
179
180          if evaluations_used >= budget:
181              break
182
183          # Sort population by fitness
184          sorted_indices = np.argsort(fitnesses)
185          sorted_population = [population[i] for i in sorted_indices]
186          sorted_fitnesses = [fitnesses[i] for i in sorted_indices]
187
188          # Select parents
189          parents = sorted_population[:mu]
190
191          # Update mean
192          old_mean = mean
193          mean_update = np.sum([weights[i] * (parents[i] - old_mean) for i in range(mu)],
                  axis=0) / initial_sigma
194          mean = old_mean + initial_sigma * mean_update
195
196          # Update evolution paths
197          ps_new = (1 - c_sigma) * ps + np.sqrt(c_sigma * (2 - c_sigma) * mu_eff) * (B @
                  mean_update)
198          ps = ps_new
199
200          hsig = np.linalg.norm(ps) / np.sqrt(1 - (1 - c_sigma)**(2 * evaluations_used /
                  population_size)) / np.sqrt(dimension) < (1.4 + 2 / (dimension + 1))
201
202          pc_new = (1 - c_c) * pc + hsig * np.sqrt(c_c * (2 - c_c) * mu_eff) * mean_update
203          pc = pc_new
204
205          # Update covariance matrix C
```

```python
206          C_old = C
207          C_rank_one = np.outer(pc, pc) if dimension > 1 else np.array([[pc[0] * pc[0]]])
208
209          C_rank_mu = np.sum([weights[i] * np.outer((parents[i] - old_mean) / initial_sigma,
                     (parents[i] - old_mean) / initial_sigma) for i in range(mu)], axis=0)
210
211          C = (1 - c_1 - c_mu) * C_old + c_1 * C_rank_one + c_mu * C_rank_mu
212
213          # Ensure C is symmetric and positive definite (numerical stability)
214          C = (C + C.T) / 2.0 + 1e-10 * np.eye(dimension) # Add small diagonal to ensure
                     positive definiteness
215
216          # Update sigma
217          initial_sigma = initial_sigma * np.exp((c_sigma / d_sigma) * (np.linalg.norm(ps) /
                     np.sqrt(dimension) - 1))
218
219          # Eigendecomposition of C for B and D
220          # Re-evaluate B and D periodically to save computation.
221          # This is typically done if the number of evaluations is a multiple of some factor
                     ,
222          # or every few generations. For simplicity and robustness, we do it every
                     generation.
223          # Note: np.linalg.eigh returns eigenvalues in ascending order.
224          D_squared, B = np.linalg.eigh(C)
225          D = np.sqrt(np.maximum(1e-10, D_squared)) # Ensure strictly positive eigenvalues
                     for numerical stability
226
227      # Fallback if everything failed/returned inf
228      if best_x is None or not np.isfinite(best_value):
229          best_x = _clip_to_bounds((lower + upper) / 2.0, lower, upper)
230          best_value = _evaluate_safe(problem, best_x)
231          if not np.isfinite(best_value):
232              best_value = float("inf")
233
234      return best_x.tolist(), float(best_value), evaluations_used
235
236
237  # EVOLVE-BLOCK-END
238
239
240  def run_search_entry(problem, budget: int = 1000, seed: int | None = None):
241      """
242      Thin wrapper kept outside the evolve block in case the block is replaced.
243      """
244      return run_search(problem, budget=budget, seed=seed)
245
246
247  if __name__ == "__main__":
248      # Smoke test with a trivial sphere if optunahub is available
249      try:
250          import optunahub
251
252          bbob = optunahub.load_module("benchmarks/bbob")
253          test_problem = bbob.Problem(function_id=1, dimension=3, instance_id=1)
254          x, value, used = run_search(test_problem, budget=100, seed=0)
255          print(f"Best value {value:.4e} after {used} evals at x={x}")
256      except Exception as exc:
257          print(f"Smoke test skipped ({exc})")
```

