# OpenReview forum: "DeltaEvolve: Accelerating Scientific Discovery through Momentum-Driven Evolution"
_ICML.cc/2026/Conference — ICML 2026 regular_

### Official Review · Reviewer_S7L1 · 2026-02-24

**Soundness:** 2
**Presentation:** 2
**Significance:** 2
**Originality:** 2
**Overall Recommendation:** 4
**Confidence:** 4

**Summary:**

This paper presents DeltaEvolve, a momentum-driven evolutionary framework to accelerate automated scientific discovery while addressing the context inefficiencies of prior methods. The authors find that current methods rely on full-code histories, which consume excessive tokens and obscure the core logic driving performance gains.
The paper proposes replacing static code snapshots with semantic deltas in evolutionary coding agents. These deltas explicitly capture "how" and "why" modifications affect performance, acting as a directional "momentum" signal. To optimize context usage, DeltaEvolve utilizes a multi-level database (summaries, detailed plans, and full code) paired with a progressive disclosure that provides granular details only for the most relevant nodes.
Evaluated across five domains, DeltaEvolve consistently discovers superior solutions while reducing token consumption by an average of 36.79%.

**Compliance With Llm Reviewing Policy:**

Affirmed.

**Final Justification:**

My concerns have been addressed and I have raised the rating accordingly.

**Key Questions For Authors:**

- Are the LLM-generated semantic deltas reliable? How do you validate that?

- What motivates the sampler design? The progressive rendering mechanism appears heuristic.

**Limitations:**

The limitations of this work are not adequately discussed.

**Strengths And Weaknesses:**

*Strengths*

The framework achieves substantial token reduction by replacing redundant full-code snapshots with structured semantic deltas.

The three-layer representation and progressive disclosure mechanism are well-designed and conceptually elegant.

Experiments are conducted across five diverse domains using state-of-the-art proprietary models.

*Weaknesses*

The motivation for involving the EM view appears unclear. Context engineering is already well-established, and the influence of context on generation is inherent to how LLMs model probability distributions. The claim that "context plays a key role in updating the search distribution and guiding evolution" does not seem to require EM formulation for validation. I do not see the value of dedicating so much space to this.

The concept of verbal gradients (functionally equivalent to the proposed semantic deltas) has been explored in prior work [1,2], yet this connection is not discussed.

The experimental results appear brittle. The number of runs performed and the variance across runs are not reported. It would be ideal for the evolution curves (with std plotted) included in the main content rather than relegated to supplementary materials.


[1]  Pryzant, R., Iter, D., Li, J., Lee, Y., Zhu, C., & Zeng, M. (2023). Automatic prompt optimization with “gradient descent” and beam search. In Proceedings of the 2023 conference on empirical methods in natural language processing (pp. 7957-7968).

[2] Ye, H., Wang, J., Cao, Z., Berto, F., Hua, C., Kim, H., ... & Song, G. (2024). Reevo: Large language models as hyper-heuristics with reflective evolution. Advances in neural information processing systems, 37, 43571-43608.

---

> ### Author Rebuttal · Authors · 2026-03-30
>
> > **Q1: The EM interpretation.**
>
> A1: We thank the reviewer for this point. We clarify that our EM framework serves three purposes: (1) **new perspective**—to our knowledge, no prior work has analyzed LLM-driven evolutionary systems through the lens of EM; (2) **unified view**—existing systems (AlphaEvolve, FunSearch, Greedy Refine) can all be understood as instantiations of the same E-step/M-step alternation, differing only in their context update policy; and (3) **principled motivation**—this unified view directly exposes the M-step context representation as the key design bottleneck, providing a principled rationale for DeltaEvolve's semantic delta design. We agree that context influences generation inherently, but the EM framing makes this implicit relationship explicit and actionable, bridging the gap from observation to systematic design.
>
> > **Q2: Connection to verbal gradients.**
>
> A2: We thank the reviewer for pointing out these references. We apologize for the oversight and will cite and discuss them in the revision. That said, we respectfully argue that semantic deltas and textual gradients are fundamentally different in multiple aspects:
>
> 1. **Different optimization targets.** ProTeGi optimizes short prompt text (a few sentences), while DeltaEvolve operates on full programs (hundreds of lines of code). These are very different optimization landscapes.
> 2. **Different lifespans.** ProTeGi's gradients are ephemeral—generated, used once to edit the current prompt, then discarded. DeltaEvolve's deltas are persistently stored in a memory bank and reused across future iterations, forming an accumulated momentum signal.
> 3. **Different roles.** ProTeGi's gradients tell the LLM "what is wrong with the current prompt." DeltaEvolve's deltas tell the LLM "what code modifications have worked in the past and why," serving as transferable inspiration for future mutations rather than error correction.
>
> For comparison with ReEvo, we refer the reviewer to `Reviewer [Hz7c] Q1/A1`.
>
> > **Q3: Variance and evolution curves.**
>
> A3: We thank the reviewer for this suggestion. Our reported results are the best scores across 3 independent runs. We follow the convention in evolutionary program synthesis, where the primary metric of interest is the best achievable solution rather than the average, as the goal is to discover a single high-quality program.
>
> That said, we provide all individual runs and summary statistics below for the black-box optimization task:
>
> | Method | Run 1 | Run 2 | Run 3 | Mean | Std |
> |--------|-------|-------|-------|------|-----|
> | AlphaEvolve | 2.6415 | 2.5194 | 2.5604 | 2.5738 | 0.0507 |
> | DeltaEvolve  | 2.7297 | 2.6843 | 2.7280 | 2.7140 | 0.0210 |
>
> DeltaEvolve consistently outperforms AlphaEvolve across all three runs, with a higher mean and lower variance, indicating both stronger performance and greater stability.
>
> Regarding evolution curves: we agree they are informative and will move them from the appendix to the main text in the revised manuscript.
>
> > **Q4: Reliable of delta.**
>
> A4: We thank the reviewer for this question. To address this concern, we conducted an **offline analysis** using an independent judge model (Claude Opus 4.6) to evaluate the alignment between delta plans and generated code. Please refer to `Reviewer[SqGV] Q3/A3` for full methodology details.
>
> Based on the results, we find that: (1) the semantic delta is highly accurate to the generated solutions, with a delta_fidelity of 99.03%; (2) the misalignment rate is very small, with nearly 98% of all major code modifications covered by at least one delta claim. We therefore believe the delta plan summary and details are well-aligned with the generated programs and do not introduce meaningful misalignment risk during evolution.
>
> > **Q5: Motivation for progressive rendering.**
>
> A5: We thank the reviewer for this question. Our progressive rendering is motivated by design pattern in agent memory management with core principle *show what exists and its retrieval cost first, and let the agent decide what to fetch based on relevance and need.* Specifically, our design principle is that the closer a node is to the current mutation, the more detail it needs. The parent node requires Level 3 (full code) because new code must be generated by directly modifying it—abstract descriptions alone cannot support valid code editing. Recent elite and diverse nodes use Level 2 (delta plan details) because their concrete logic changes are most likely to transfer to the current parent. Distant nodes use Level 1 (delta summary) because details lose relevance over evolutionary time—only high-level strategic direction remains valuable at that distance. This design is empirically validated by our token efficiency results (Table 2), where DeltaEvolve achieves superior scores with significantly fewer tokens. We will incorporate this discssion in the revision.

---

> > ### Author Rebuttal · Reviewer_S7L1 · 2026-04-03
> >
> > My concerns have been addressed and I have raised the rating accordingly.

---

> > > ### Author Response · Authors · 2026-04-06
> > >
> > > Thank you for raising the score. We are glad to hear that your concerns have been addressed.
> > >
> > > Please feel free to reach out if you have any further questions or need additional clarification. We truly appreciate your time and feedback.

---

### Official Review · Reviewer_HU5t · 2026-03-10

**Soundness:** 3
**Presentation:** 1
**Significance:** 4
**Originality:** 3
**Overall Recommendation:** 4
**Confidence:** 3

**Summary:**

This paper tackles the problem of evolving programs for scientific discovery while minimizing context window size. The paper argues that existing approaches utilize inefficient full code trajectories, which provide sub-optimal signaling and inflate the context window. Instead, the proposed approach utilizes a semantic delta between update steps to capture meaningful changes to approach and drive better evolution outcomes. The main contributions are the re-framing of evolutionary code generation under the traditional EM (Expectation Maximization) framework, effective implementation of the semantic delta signal, and empirical experiments comparing AlphaEvolve, greedy refinement, and parallel decoding to the proposed DeltaEvolve method.

**Compliance With Llm Reviewing Policy:**

Affirmed.

**Key Questions For Authors:**

(1) In the sub-section Node Selection and step 3 (Diverse Nodes), how are the text embeddings calculated? Are these text embeddings sufficient to capture potentially nuanced logic that may be contained within the nodes? What are the average distances between text embeddings as the number of retained nodes grows? Demonstrating the text embeddings capture appropriate semantics would improve technical soundness of the paper.

(2) How was the probability of selecting lower reward nodes chosen? How sensitive is the evolution trajectory to this probability? Showing robustness to probability selection would improve the significance of this paper.

(3) How accurate is the delta plan summary and delta plan details to the generated solutions? Have there been instances of misalignment between the output delta plan summary and the content of the proposed program? The evolutionary trajectory seems particularly sensitive to such misalignment when large logical changes are made between iterations. Showing that the delta plan summary and details don't suffer from misalignment would improve the significance of this paper.

(4) How were the model families for evaluation chosen? Showing the model family selection is not biased would improve the technical soundness of the paper.

**Limitations:**

Yes

**Strengths And Weaknesses:**

The paper has moderate technical soundness. The casting of evolutionary program generation into the EM framework (and the role of the context as the sole parameterization of latents) is interesting, and the need for more efficient stepwise updates is justified. However, the extraction of semantic deltas requires more careful exposition and is unclear in sections. Experimental design is mostly sound but limited in scope; given the approach is applicable to a wide variety of problems (as noted by the authors) the paper would be improved by utilizing some of the medium to large datasets available within those problem domains to support the utility of the semantic delta signal. The evaluation metrics and research questions are appropriate and clear.

The paper has low presentation quality. While certain sections (sections 2, 3) are clear and well-motivated, sections containing key technical details (section 4) require either concrete examples or better diagrams to justify their claims. At the moment, many diagram elements are too generic to generate insight, and appendix figures like figure 12,13 would build useful intuition if used early in the main paper. Additionally, the authors repeatedly use non-standard terminology early in the paper when a formal definition or sufficient motivation has not yet been developed.

The paper has high significance. Utilization of evolutionary methods for scientific law discovery and program generation often suffers from context bloat, and reasoning models can further drive up token usage and potential environmental impacts.

The paper has low to moderate originality. While utilizing the EM framework to describe evolutionary code generation or scientific law discovery is not a new concept, the utilization of context windows within an LLM generator in this framework is interesting, and certain experiments (like section 3.3 - table 1) help illuminate evolutionary trajectories. Furthermore, the idea of capturing semantic change along with a directional gradient signal is useful but well explored territory in evolution based approaches. The extraction of semantic delta and the relative effect on token usage are novel contributions.

---

> ### Author Rebuttal · Authors · 2026-03-30
>
> > **Q1: About text embeddings.**
>
> A1: We thank the reviewer for this insightful question. First, we use OpenAI's `text-embedding-3-large` to compute embeddings of the full program code. Diverse nodes are selected based on maximum embedding distance from the parent. Second, text embeddings are sufficient to capture nuanced logic: modern large-scale embedding models are trained on extensive code corpora and capture semantic similarity well beyond surface-level text overlap. This capability is widely validated in code retrieval, clustering, and recommendation systems.
>
> To more rigorously measure program diversity in the embedding space, we computed two metrics: (1) the pairwise cosine similarity matrix across all retained programs, and (2) the average cosine similarity as iterations progress. Results can be found in Figure 3 of https://anonymous.4open.science/r/deltaevolve_rebuttal-3C4E/DeltaEvolve_rebuttal.pdf. The pairwise cosine similarity ranges from 0.408 to 1.0, and the similarity matrix confirms that embeddings exhibit meaningful diversity without collapsing to uniformly high similarity. The average cosine similarity initially fluctuates and then stabilizes around 0.475, demonstrating that the population maintains sustained diversity throughout the evolutionary process. We would add these analysis in our revision.
>
> **Q2: Lower reward nodes chosen and robustness.**
>
> A2: We thank the reviewer for this question. Our parent selection policy follows the AlphaEvolve baseline, balancing exploitation with exploration: at each iteration, with probability *Exploit Ratio*, a parent is selected from the elite archive of top-performing solutions; otherwise, a node is selected uniformly at random from the island population, which naturally includes lower-reward nodes. The exploit ratio is reported in Table 4.
>
> Regarding sensitivity: we conducted a robustness analysis on the black-box optimization task (100 programs, DeltaEvolve based on ShinkaEvolve) by varying the exploit ratio:
>
> | Setting | Exploit Ratio | Best Score |
> |---------|--------------|------------|
> | Baseline | 0.7 | 2.9097 |
> | Variant 1 | 0.6 | 2.9095 |
> | Variant 2 | 0.8 | 2.9093 |
> | Extreme 1 | 1.0 | 2.8396 |
> | Extreme 2 | 0.0 | 2.7911 |
>
> The results show that DeltaEvolve is robust to moderate variations in the exploit ratio: shifting ±0.1 from the baseline yields nearly identical best scores (2.9093–2.9097). Performance degrades only under extreme settings that eliminate either exploration or exploitation entirely, confirming that a balanced selection policy is beneficial but the exact ratio is not a sensitive hyperparameter.
>
> > **Q3: Fidelity of deltas.**
>
> A3: We thank the reviewer for this question. To address this concern, we conducted an **offline analysis** using an independent judge model (Claude Opus 4.6) to evaluate the alignment between delta plans and generated code. Please refer to `Reviewer[SqGV] Q3/A3` for full methodology details.
>
> Based on the results, we find that: (1) the semantic delta is highly accurate to the generated solutions, with a delta_fidelity of 99.03%; (2) the misalignment rate is very small, with nearly 98% of all major code modifications covered by at least one delta claim. We therefore believe the delta plan summary and details are well-aligned with the generated programs and do not introduce meaningful misalignment risk during evolution.
>
> > **Q4: Model families chosen.**
>
> A4: We thank the reviewer for this question. Our experiments employ two model selection strategies: (1) fixed probabilistic routing (e.g. OpenEvolve), where models are sampled at predefined ratios (e.g., 0.8 for gpt-5-mini and 0.2 for o3-mini); and (2) adaptive LLM routing (e.g., ShinkaEvolve's UCB1-based strategy), which dynamically selects the best-suited model based on relative improvement throughout evolution. The semantic delta representation improves performance under both strategies, demonstrating that our gains are not tied to a specific model selection scheme. We refer the reviewer to the table in `Reviewer[Hz7c] Q1/A1` for detailed results.

---

> > ### Author Rebuttal · Reviewer_HU5t · 2026-04-06
> >
> > (a) The authors provide a sufficiently detailed and rigorous response to the critiques raised. The additional table provided for response to Q2 and the alignment analysis done for Q3 further solidify the technical soundness of the paper.

---

> > > ### Author Response · Authors · 2026-04-07
> > >
> > > Thank you for your feedback. We are glad to hear that your concerns have been fully addressed.
> > >
> > > We appreciate your time and thoughtful evaluation. If you have any further questions or need additional clarification, we would be happy to address.

---

### Official Review · Reviewer_SqGV · 2026-03-12

**Soundness:** 3
**Presentation:** 3
**Significance:** 3
**Originality:** 3
**Overall Recommendation:** 4
**Confidence:** 3

**Summary:**

The paper proposes DeltaEvolve, an LLM-driven evolutionary framework for program-based scientific discovery that replaces full-code historical contexts with structured “semantic deltas” describing what changed and why it affected performance. The authors formalize evolution as an EM-like process in which the E-step samples candidates and the M-step updates the context policy; they argue that full-code histories are a suboptimal M-step and introduce a multi-level database plus a progressive disclosure sampler to maximize informativeness under token budgets. Experiments on five domains suggest DeltaEvolve improves best-found solutions while reducing token consumption by about 36.8% on average compared to full-code baselines such as AlphaEvolve/OpenEvolve.

**Compliance With Llm Reviewing Policy:**

Affirmed.

**Final Justification:**

While the authors' rebuttal has successfully addressed my initial concerns, I still consider this manuscript to be a borderline submission.

**Key Questions For Authors:**

See Weakness.

**Limitations:**

yes

**Strengths And Weaknesses:**

## Strengths
-	Reframing iterative LLM agents for program evolution as an EM-style optimization that emphasizes context construction as the key “latent” lever is a useful conceptualization for the community.
-	The semantic delta representation (Level 1 summaries and Level 2 structured plans) offers a principled alternative to full-code context, with a momentum-like intuition that aggregates directional improvements.
-	Evaluation spans five distinct domains (black-box optimization, packing, symbolic regression, PDE solver, and kernel optimization), and includes results with two LLM families and an ensemble setting to probe robustness.
-	The EM framing and the momentum analogy are explained intuitively, helping position the contribution.

## Weakness
-	The EM interpretation is largely analogical rather than formal: there is no explicit likelihood/objective whose E/M steps are being optimized, and “context as latent variable” is not accompanied by guarantees or convergence arguments.
-	The “momentum” analogy is suggestive but not grounded by measurements (e.g., quantifying reuse/transfer of deltas across branches or showing accelerated progress due to accumulated deltas).
-	Fidelity of deltas: all delta levels are co-generated by the same LLM that wrote the code. There is no verification that deltas faithfully reflect AST/code diffs, nor checks against hallucinated causal explanations.
-	Fairness of baselines is under-specified: the proposed sampler uses MAP-Elites and progressive rendering; it is unclear whether AlphaEvolve/OpenEvolve baselines receive the same orchestration (except for the representation of history), which could confound gains.

---

> ### Author Rebuttal · Authors · 2026-03-30
>
> > **Q1: The EM interpretation.**
>
> A1: We thank the reviewer for this point. We want to clarify that our EM framework serves three purposes: (1) it offers a **new perspective**—to our knowledge, no prior work has analyzed LLM-driven evolutionary systems through the lens of EM; (2) it provides a **unified view**—existing systems (AlphaEvolve, FunSearch, Greedy Refine) can all be understood as instantiations of the same E-step/M-step alternation, differing only in their context update policy; and (3) it **motivates** our core contribution—this unified view directly exposes the M-step context representation as the key design bottleneck, providing a principled rationale for DeltaEvolve's semantic delta design.
>
> Regarding formal convergence guarantees: this remains an open challenge for the entire problem class. No existing LLM-driven evolutionary system—including AlphaEvolve and FunSearch—provides such analysis. We agree with the reviewer that developing a formal convergence theory for LLM-based evolutionary systems is an important direction. We believe EM provides a promising framework for theoretical analysis, but we leave this to future work.
>
> > **Q2: The "momentum" analogy.**
>
> A2: We thank the reviewer for this suggestion. We agree that empirically grounding the momentum analogy would strengthen the paper. Directly quantifying reuse/transfer of deltas across branches is difficult because the LLM does not copy delta inspirations—measuring semantic similarity between context deltas and resulting deltas would not reliably capture the **causal effect**. Instead, we adopt a perturbation-based approach to isolate the contribution of accumulated deltas. We compare three settings:
>
> - **No-Delta**: parent code only, with all semantic deltas removed in context.
> - **Degraded-Delta**: parent code + accumulated semantic delta from score-dropping branches.
> - **DeltaEvolve (Ours)**: parent code + accumulated semantic delta from high-performing + diverse branches.
>
> | Setting | Best Score | Token Consumption |
> | :--- | ---: | ---: |
> | No-Delta | 2.6740 | 89146 |
> | Degraded-Delta | 2.6203 | 928265 |
> | **DeltaEvolve (Ours)** | **2.9097** | **921364** |
>
> Both No-Delta and Degraded-Delta lead to substantially worse performance, confirming that the quality and accumulation of historical deltas—not merely the presence of additional context—drives improvement. This provides causal evidence for the momentum effect described in the paper. We will include this experiment in the revised manuscript.
>
> > **Q3: Fidelity of deltas.**
>
> A3: We thank the reviewer for this concern. The reason we co-generate deltas with the code—rather than introducing a separate verification step during evolution—is to avoid additional LLM calls and token cost, which would contradict our core goal of token efficiency. To confirm that this design choice does not compromise fidelity, we conducted an **offline analysis** on the evolution trajectories. Specifically, we extracted every modification block from each semantic delta along with the corresponding code diff, and submitted each (claim, diff) pair to an independent judge model (Claude Opus 4.6) for evaluation. We measure two complementary metrics: **delta_fidelity**, the fraction of individual delta modification judged as supported by the actual diff; and **delta_coverage**, the fraction of major code changes that are covered by at least one delta claim.
>
> | Metric | Value | 95% CI |
> | :--- | ---: | ---: |
> | delta_fidelity | 0.9903 | [0.9838, 0.9957] |
> | delta_coverage | 0.9798 | [0.9597, 0.9960] |
>
> The results show that 99% of delta claims are faithful to the actual code changes, and that deltas cover 98% of all major code modifications. These results confirm that co-generated deltas are empirically faithful, validating our design choice of prioritizing token efficiency without sacrificing representation accuracy.
>
> **Q4: Fairness of baselines.**
>
> A4: We thank the reviewer for raising this concern. The comparison is fair: both DeltaEvolve and AlphaEvolve employ the same MAP-Elites diversity selection mechanism and identical population management strategies. The only difference is the mutation context representation—full code (AlphaEvolve) vs. semantic deltas (ours). Progressive rendering is designed to reduce token consumption rather than improve solution quality, and is a natural consequence of having multi-level deltas—it is not applicable to systems that condition on full code only.
>
> More broadly, semantic delta is **orthogonal** to the search strategy. It operates at the mutation representation level without modifying population management, parent selection, or any other orchestration component. This means it can be plugged into any existing evolutionary system that currently uses full code as LLM context without altering their original design. For instance, see `Reviewer[Hz7c] Q1/A1` for new results on ShinkaEvolve. We will incorporate the above discussion in the revision.

---

> > ### Author Rebuttal · Reviewer_SqGV · 2026-04-02
> >
> > While the authors' response and the additional experiments successfully resolve the specific issues I raised, my overall evaluation of the manuscript remains unchanged. Therefore, I decide to maintain my score.

---

> > > ### Author Response · Authors · 2026-04-06
> > >
> > > Thank you for your feedback. We are glad to hear that your concerns have been fully addressed.
> > >
> > > We appreciate your time and thoughtful evaluation. If you have any further questions or need additional clarification, we would be happy to answer.

---

### Official Review · Reviewer_Hz7c · 2026-03-13

**Soundness:** 3
**Presentation:** 4
**Significance:** 3
**Originality:** 2
**Overall Recommendation:** 4
**Confidence:** 4

**Summary:**

This work proposes DeltaEvolve, an LLM-based framework for evolutionary program search that replaces full-code history in the context window with structured "semantic deltas", which are natural language descriptions of what changed between successive programs and why. The authors show empirically that selection policy dominates scalar feedback and use different levels of detail for delta messages based on recency. Experiments across five scientific domains show DeltaEvolve achieves comparable or better accuracy than AlphaEvolve and reduces token consumption.

**Compliance With Llm Reviewing Policy:**

Affirmed.

**Final Justification:**

Rebuttal could address some of my concerns. I keep my positive score and support for the paper, but my general view on the paper's originality and idea remains the same, and thus not increasing my score.

**Key Questions For Authors:**

1. How does DeltaEvolve compare against recent works such as ReEvo, EoH, ShinkaEvolve, etc. For example, ReEvo uses short-term and long-term reflections that are conceptually very similar to L1/L2 deltas. Can the authors clarify how DeltaEvolve's contribution is distinct from these, and ideally provide empirical comparisons?

2. The quality gains with GPT models are modest while Gemini shows much larger improvements. Is there an explanation for this discrepancy?

3. The token savings could partly be explained by programs becoming longer over iterations (making full-code contexts more expensive in absolute terms), independently of the delta representation. Can the authors provide a per-iteration token breakdown or control for program length to isolate the contribution of the delta format itself?

4. Since the AlphaEvolve baseline is reproduced via OpenEvolve, how sensitive are the reported comparisons to the fidelity of this reproduction?

**Limitations:**

yes

**Strengths And Weaknesses:**

Strengths:
* The selection policy vs. scalar feedback study provides a good motivation to the method
* The semantic delta concept is simple yet practical, reducing context length and improving performance.
* The results show consistent improvement over AlphaEvolve; The paper is well written.

Weaknesses:
* Several recent works improve upon AlphaEvolve and some of them even include natural language concepts and similar delta messages (directional guidance), and decomposition of the solution, but none of those are provided as baselines for experiments and discussed well in the experiments/discussion
* The various levels of Delta sound reasonable but not supported empirically (e.g., by ablation)

---

> ### Author Rebuttal · Authors · 2026-03-29
>
> > **Q1: Comparison with ReEvo, EoH, ShinkaEvolve.**
>
> A1: We thank the reviewer for highlighting the conceptual connections to recent works. We would cite ReEvo and EoH and clarify DeltaEvolve's distinction from two perspectives:
>
> - **Context for generating child programs.** All of these methods still condition on full code (often from multiple historical nodes) as the primary source of inspiration. DeltaEvolve instead conditions on a semantic delta—a natural-language description of the intended plan change from parent to child—so the LLM generates new code guided by "what changed" rather than re-ingesting entire code histories.
>
> - **Role of natural-language semantics.** ReEvo's short-/long-term reflections and ShinkaEvolve's scratchpad are forms of reflection, typically distilled by an additional LLM call after code evaluation. In contrast, DeltaEvolve's semantic delta is not a reflection mechanism—it directly encodes the plan change from parent to child and is produced jointly during code generation, requiring no extra LLM calls and thus improving efficiency. EoH also employs natural-language plans, but each plan describes a single program in isolation rather than modeling the transition between two programs, missing the structured parent→child change signal that our delta representation captures.
>
> **Empirical comparison.** To directly compare these frameworks, we evaluate on our black-box optimization problem. For each method, we generate 100 programs per run, repeat 3 times, and report the best score. Since our semantic delta method is composable with existing search frameworks, we evaluate it on both AlphaEvolve (OpenEvolve) and ShinkaEvolve:
>
> | Method | Best Score ↑ | Token Consumption ↓|
> |---|---|---|
> | EoH | 2.4649 | 480,876 |
> | ReEvo | 2.5979 | 1,060,408 |
> | AlphaEvolve | 2.6415 | 1,852,841 |
> | **DeltaEvolve (Based on AlphaEvolve)** | **2.7297** | **1,390,709** |
> | ShinkaEvolve | 2.7076 | 2,230,726 |
> | **DeltaEvolve (Based on ShinkaEvolve)** | **2.9097** | **921,364** |
>
> Our semantic delta consistently improves score and reduces token cost across different frameworks.
>
> > **Q2: Discrepancy in gains across GPT and Gemini models.**
>
> A2: We thank the reviewer for this insightful observation. We clarify that the discrepancy is heavily task-dependent rather than model-dependent. A counter-example is Symbolic Regression (Table 2), where GPT shows substantial improvement (3.2657→3.4174) while Gemini's gain is negligible (3.2174→3.2198). This is expected because different LLMs have varying strengths across problem domains—each model's pretraining data and reasoning capabilities make it better suited for certain types of code transformations than others.
>
> A more principled solution is adaptive LLM routing (e.g., ShinkaEvolve's UCB1-based strategy), which dynamically selects the best-suited model based on relative improvement throughout evolution. Since our semantic delta is orthogonal to model selection, it composes naturally with such routing. Our results on ShinkaEvolve (Table in Q1) confirm that combining both yields further gains.
>
> > **Q3: Isolating the contribution of the delta format from program length growth.**
>
> A3: We thank the reviewer for this thoughtful concern. To address it, we plot program length (in tokens) across iterations for both DeltaEvolve and the AlphaEvolve baseline. As shown in Figure 1 in https://anonymous.4open.science/r/deltaevolve_rebuttal-3C4E/DeltaEvolve_rebuttal.pdf, program length does not monotonically increase over iterations in either setting—it varies around a stable range (~1500–3500 tokens) without a clear upward trend. This is expected: evolutionary improvements typically refine program logic (e.g., better heuristics, smarter branching) rather than appending more code, so higher fitness does not imply longer programs. Further, we provide the per-iteration comparison in Figure 2 and find token saving is consistent across all iterations. This indicates the inherent efficiency of the delta representation itself: encoding "what to change" is consistently more compact than re-transmitting the full program, regardless of program length.
>
> > **Q4: Sensitivity to OpenEvolve reproduction fidelity.**
>
> A4: We address this concern from two angles. First, regarding reproduction fidelity: we have thoroughly inspected OpenEvolve's core modules (LLM-based generation, evaluation, and population management) to ensure faithful alignment with the original AlphaEvolve design. Second, and more importantly, our semantic delta representation is **orthogonal** to the choice of evolutionary framework—replacing full-code context with semantic deltas is a simple, general-purpose modification that can be applied to any evolve system with minimal effort. To demonstrate this, we further integrate the delta format into ShinkaEvolve (see Table in Q1) and get consistent improvements. This confirms that the gains from semantic delta are robust across evolutionary frameworks.

---

> > ### Author Rebuttal · Reviewer_Hz7c · 2026-04-04
> >
> > Thanks for the detailed responses. My concerns are addressed. I keep my positive score and support for the paper, but my general view on the paper's originality and idea remains the same, and thus not increasing my score.

---

> > > ### Author Response · Authors · 2026-04-06
> > >
> > > Thank you for your feedback. We are glad to hear that your concerns have been fully addressed.
> > >
> > > We appreciate your time and thoughtful evaluation. If you have any further questions on the originality of our work, we would be happy to answer.

---

### Decision · Program_Chairs · 2026-04-30

**Decision:**

Accept (regular)

**Comment:**

This paper proposes DeltaEvolve, a framework for evolutionary program search that replaces full-code histories with structured semantic deltas to improve context efficiency.

Reviewers find the idea simple and practical, with consistent empirical improvements and clear reductions in token usage across multiple domains. Following the rebuttal, most technical concerns have been addressed, and the overall assessment remains positive.

However, reviewers continue to express reservations about the level of novelty, the strength of the conceptual framing (e.g., EM interpretation), and the rigor of certain experimental choices and baselines. These concerns limit the overall impact but do not undermine the core contribution.

I recommend Accept.